# Cdh2, a downstream target of Hes7, regulates somitogenesis by supporting FGF signalling

Xueqi Jia[1,2,*], Akihiro Isomura[1,3,4,5,*] and Ryoichiro Kageyama[1,4,*,‡]

## ABSTRACT

In the segmentation clock, Hes7 expression oscillates synchronously in the presomitic mesoderm (PSM), regulating periodic somite formation. Despite intensive studies, the whole regulatory gene networks of the segmentation clock remain to be analysed. To identify the direct target genes of Hes7, we performed chromatin immunoprecipitation with sequencing analysis using an anti-Hes7 antibody and knocked out the identified genes from mouse embryonic stem cells carrying a Hes7 reporter. These cells were induced to differentiate into PSM-like tissue, and live imaging of Hes7 oscillations was conducted. Among the tested genes, *Cdh2* knockout resulted in downregulation of fibroblast growth factor (FGF) signalling and premature cessation of Hes7 oscillations. Conversely, *Cdh2* overexpression led to upregulation of FGF signalling and prolonged Hes7 oscillations. Whereas *Cdh2* mRNA showed dynamic expression through repression by Hes7 oscillations, Cdh2 protein exhibited a rather steady gradient with higher levels in the posterior PSM and lower levels in the anterior PSM. Thus, Hes7-controlled Cdh2 regulates FGF signalling, leading to the proper maintenance of Hes7 oscillations, suggesting that the interplay between Hes7 and Cdh2 governs the timing mechanism of PSM differentiation.

KEY WORDS: Cdh2, Hes7, Oscillatory expression, Presomitic mesoderm, Segmentation clock, Mouse

## INTRODUCTION

Embryos develop from single cells into mature organisms through dramatic transformations that are tightly regulated by temporal and spatial cues. How each cell interprets these signals to determine when and where to differentiate during morphogenesis remains an intriguing question. Among various developmental processes, somitogenesis in vertebrate embryos is a prime example. During this process, presomitic mesoderm (PSM) cells mature along the posterior-anterior axis and differentiate in a sequential manner to form somites, which are segmented blocks of mesodermal tissue that later give rise to metameric structures, such as vertebrae, ribs

[1]RIKEN Center for Brain Science, Wako 351-0198, Japan. [2]Kyoto University Graduate School of Biostudies, Kyoto 606-8501, Japan. [3]Institute for Frontier Life and Medical Sciences, Kyoto University, Kyoto 606-8507, Japan. [4]Institute for Integrated Cell-Material Sciences, Kyoto University, Kyoto 606-8501, Japan. [5]Japan Science and Technology Agency, PRESTO, Saitama 332-0012, Japan. *Present address: RIKEN Center for Biosystems Dynamics Research, Kobe 650-0047, Japan.

‡Author for correspondence (ryoichiro.kageyama@riken.jp)

 R.K., 0000-0002-5985-1120

and skeletal muscles. The segmentation clock, a network of molecular oscillators primarily governed by signalling pathways such as Notch, Wnt and fibroblast growth factor (FGF) (Bessho et al., 2001a,b; Aulehla et al., 2003; Niwa et al., 2007; Wahl et al., 2007), regulates rhythmic segmentation of the PSM, controlling periodic somite formation. Another important component in this process is the gradient distribution of Wnt and FGF pathway factors, such as Wnt3A and FGF4/8, which regulate the differentiation front (Dubrulle et al., 2001; Sawada et al., 2001; Delfini et al., 2005; Aulehla et al., 2008). Extensive evidence from genetic experiments and chemical interventions has led to the development of evolving models that aim to elucidate the relationship between the segmentation clock and these gradients (McDaniel et al., 2024). This growing body of research highlights the need to explore further the hierarchy and interactions of these regulatory mechanisms in somitogenesis (Isomura and Kageyama, 2025).

Among the core components of the segmentation clock, the Hes/Her family genes, which function as transcriptional repressors downstream of the Notch signalling pathway, generate oscillator networks. In mice, a key player in this network is *Hes7*, expression of which oscillates due to a negative feedback loop (Bessho et al., 2003; Hirata et al., 2004). The loss or sustained expression of Hes7 disrupts segmentation and impairs body trunk development, suggesting that its oscillatory expression is required for these processes (Bessho et al., 2001b; Hirata et al., 2004). Another oscillator gene, lunatic fringe (*Lfng*), a key modulator of Notch signalling, is regulated by Hes7 and plays a crucial role in maintaining synchronized oscillatory Notch activity to ensure proper segmentation (Cole et al., 2002; Bessho et al., 2003; Okubo et al., 2012; Yoshioka-Kobayashi et al., 2020). Additionally, Hes7 regulates the cyclic expression of Dusp4, which is involved in oscillatory FGF/dual-phosphorylated ERK (ppERK) activity in the PSM (Niwa et al., 2011; Simsek et al., 2024). Although the Hes7-induced oscillatory networks have been well characterized, the full regulatory mechanisms and downstream effectors remain incompletely understood.

Although the functional analysis of genes in the segmentation clock has been facilitated by recent advancements in technologies, including gene editing by CRISPR and live imaging, directly testing these approaches on mouse embryos remains challenging because of the time and expense involved. The development of *in vitro* methods to induce PSM-like (iPSM) tissue from embryonic stem cells (ESCs) has provided a powerful new platform for studying somitogenesis and has expanded the experimental repertoire, enabling a deeper exploration of complex developmental questions (Chal et al., 2015; Matsumiya et al., 2018; Veenvliet et al., 2020; van den Brink et al., 2020; Isomura et al., 2025).

Here, to clarify the regulatory networks of Hes7 in the mouse PSM, we conducted chromatin immunoprecipitation with sequencing (ChIP-seq) analysis to identify a cohort of genes targeted directly by Hes7. By leveraging *in vitro* systems and gene editing, we analysed

the possibility that some of the Hes7 target genes, such as *Lfng*, may control Hes7 oscillations. We found that Hes7 regulates the expression of Cdh2, a key adhesion molecule, in PSM cells; Cdh2 is a crucial factor in the epithelial-to-mesenchymal transition and essential for proper somite structures due to its role in cell adhesion (Radice et al., 1997; Linask et al., 1998; Horikawa et al., 1999). Notably, Hes7 oscillations are prematurely dampened in *Cdh2* knockout (KO) iPSM tissue. We further examined how Cdh2 regulates Hes7 oscillations and found that the interplay between Cdh2 and FGF signalling plays an important role in the proper maintenance of Hes7 oscillations.

## RESULTS

### ChIP-seq identifies *Cdh2* as a direct downstream target of Hes7

To understand how Hes7 regulates the segmentation clock during somitogenesis, genome-wide ChIP-seq was performed in mouse embryonic day (E) 10.5 PSM using an anti-Hes7 antibody, generating a list of genes controlled directly by Hes7 (Fig. 1A). A total of 913 peaks were identified and associated with 1260 potential target genes regulated by Hes7 in the mouse PSM (Fig. S1A, Table S1). Most of the binding sites were enriched around gene promoters and near transcription start sites (Fig. S1B-D). As expected, Hes7 binding to its own promoter ranked among the top 10 peaks (Fig. 1B, Table S1), consistent with its role in generating a negative feedback loop for its own expression (Bessho et al., 2001a,b, 2003). Genes known to be downstream targets of *Hes7* and essential for the segmentation clock, such as *Lfng* and *Dusp4*, were also among the highest-ranked targets (Table S1). In addition, other candidate genes were identified, including some previously implicated in somitogenesis, e.g. *Pcdh8* and *Cdh2* (Fig. 1D) (Chal et al., 2017), genes differentially expressed in *Hes7* KO mice according to previously published microarray data, e.g. *Smoc1* and *Eno3* (Niwa et al., 2007), and genes not previously associated with this process (Table S1). Gene ontology analysis showed that the putative Hes7 target genes are involved in various developmental processes, including cell differentiation, proliferation and cell death (Fig. S1E), suggesting that Hes7 is

linked to many processes besides the segmentation clock. From the list, 20 genes that displayed >2.0-fold higher expression in *Hes7*-null mice than in *Hes7*-overexpressing mice (Niwa et al., 2007) and high expression levels in mouse and human PSM (Diaz-Cuadros et al., 2020; Matsuda et al., 2020) were selected for further analysis (Fig. S2).

To investigate whether the selected genes are important for maintaining Hes7 oscillations, each gene was knocked out using CRISPR/Cas9 technology in mouse ESCs carrying a p*Hes7*-Achilles reporter (Fig. 1A) (Isomura et al., 2025). Two or three cell lines were selected for each gene with frameshift mutations confirmed by Sanger sequencing around the target region (Fig. S3). These ESC lines were differentiated into iPSM tissue by 48-h treatment with BMP4, followed by 48-h incubation in CHIR99021 (GSK3β inhibitor)- and LDN193189 (BMP antagonist)-containing (CL) medium (see Materials and Methods). These iPSM tissues were imaged in an optimized CL medium supplemented with basic fibroblast growth factor (FGF) and BMS493, a retinoic acid inhibitor (CLBF medium) (Hubaud et al., 2017; Isomura et al., 2025), starting from 96 h post-induction. Hes7 oscillations in wild-type (WT) iPSM tissues were stable and prolonged (Movie 1, left), while they were severely disrupted in *Hes7* KO (Movie 1, right) or *Lfng* KO iPSM tissue (Fig. 1C, Fig. S4A), as observed in mutant mouse embryos (Bessho et al., 2001a,b; Yoshioka-Kobayashi et al., 2020). *Kpnb1* KO and *Smc6* KO ESCs were not viable, but all other KO ESCs for the selected candidates were viable and successfully differentiated into iPSM tissue, as confirmed by detection of p*Hes7*-Achilles activity (Figs S2 and S4). Compared to the WT cells, most KO lines showed no substantial differences in Hes7 oscillations (Fig. S4B). *Tra2b* KO iPSM tissue displayed cell apoptosis and the rapid dampening of pHes7-Achilles activity (Fig. S4A). In contrast, *Cdh2* KO iPSM tissue displayed the earlier cessation of Hes7 oscillations compared to the control tissue, while their viability was unaffected (Fig. 1E, Movie 2). Thus, among the tested genes, only *Cdh2* KO iPSM tissue showed abnormal Hes7 oscillations without apoptosis; as such, we investigated its role in Hes7 oscillations and the differentiation programmes of the PSM.

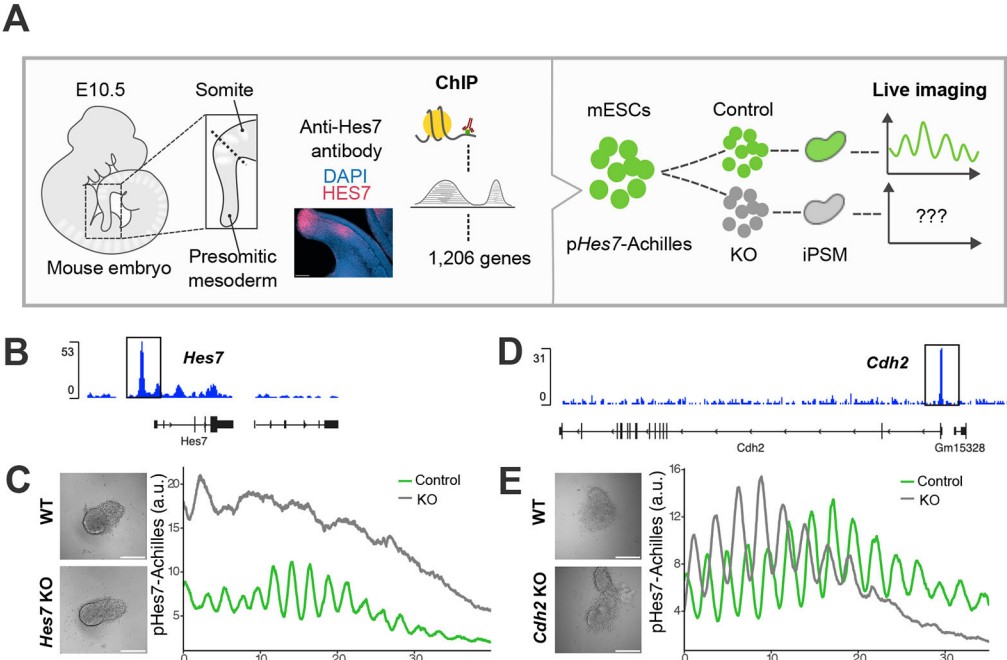

**Fig. 1. ChIP-seq identifies *Cdh2* as a direct downstream target of Hes7.** (A) Scheme of the experimental plan of the ChIP-seq analysis of Hes7 in mouse PSM cells using anti-Hes7 antibody (verified by immunostaining) and KO screening of potential candidates involved in regulating segmentation clock. (B,D) Profiles for Hes7-binding peaks around the *Hes7* (B) and *Cdh2* (D) regions visualized using the online IGV Browser. Peak enrichments are shown in blue and the gene schematic is shown below in black. (C,E) Representative brightfield images (left) of iPSM tissue plated on a fibronectin-coated dish and plots (right) of the time series of pHes7-Achilles activity of ROIs from 99 h (shown as 0 h in the graph) to approximately 140 h for *Hes7* KO (C) and for *Cdh2* KO (E). a.u., arbitrary units. Scale bars: 100 μm.

Cdh2, a member of the cadherin family involved in calcium-dependent cell–cell adhesion, plays a major role in tissue remodelling during somitogenesis (Hatta et al., 1987; Duband et al., 1987; Radice et al., 1997). Defects in somite structures have been reported in *Cdh2* KO chicken and mouse embryos, and altered cell movement has been observed in *cdh2* KO zebrafish embryos (Das et al., 2017). However, somitogenesis is initiated normally in *Cdh2* KO mouse embryos (Radice et al., 1997), and thus the role of Cdh2 in the segmentation clock remains poorly understood. The morphology of *Cdh2* KO iPSM tissue was comparable to that of WT tissue, although it was slightly more elongated than WT (Fig. 1E, Fig. S8C). In addition, synchronized oscillations were clearly observed at the beginning of imaging (Fig. 1E, Movie 3). These results suggest that altered cell–cell adhesion is not a major cause of the earlier cessation of Hes7 oscillation in *Cdh2*-KO iPSM.

## Repression of Cdh2 by Hes7 in the PSM results in transcriptional dynamics but gradient-like protein distribution

To further understand how Cdh2 is linked to segmentation clock in mouse PSM, we analysed whether Hes7 expression alters Cdh2 expression. We examined whether Hes7 represses *Cdh2* transcription in PSM cells by evaluating *Cdh2* levels in iPSM tissue overexpressing Hes7. To induce sustained Hes7 overexpression after cells commit to a PSM fate, the Tet-ON system was used, and doxycycline

was administered after 96 h of iPSM induction (Fig. 2A,B). Hes7 overexpression in iPSM samples for 24 h significantly repressed *Cdh2* mRNA (Fig. 2D), but *Tbx6* expression was only slightly affected (Fig. 2C). iPSM cells with high Hes7 expression (mCherry[+]) gathered into clusters, and immunofluorescence signals for Cdh2 protein in the mCherry[+] regions were reduced in these clusters compared to the surrounding cells (Fig. 2E). These results demonstrated that the upregulation of Hes7 in PSM cells downregulates Cdh2 level, leading to the Cdh2-low cells sorting out from the Cdh2-high populations. We next compared *Cdh2* mRNA expression between WT and *Hes7* KO iPSM tissues (Fig. 2F,G), which were embedded in a 10% Matrigel/NDiff227 solution at 96 h post-induction, forming somite-like structures in WT iPSM (Fig. 2F, arrowheads). WT iPSM tissues showed higher *Cdh2* intensity at the posterior tip, decreasing in the anterior region where *Mesp2*, a marker for the anterior PSM, was expressed (Fig. 2F,G, top). In contrast, in *Hes7* KO iPSM tissue, *Cdh2* mRNA levels remained constantly high in the *Mesp2*[+] anterior region, and the intensity gradient was flatter in *Hes7* KO iPSM tissue than in WT tissue (Fig. 2F,G, bottom). These results confirmed that Cdh2 expression was repressed by Hes7, suggesting that *Cdh2* expression could be dynamic.

To gain deeper insight into the relationship between *Hes7* and *Cdh2*, the *Cdh2* mRNA expression was examined in the PSM of WT mouse embryos by *in situ* hybridization chain reaction (HCR). This analysis revealed that *Cdh2* mRNA is highly expressed in the

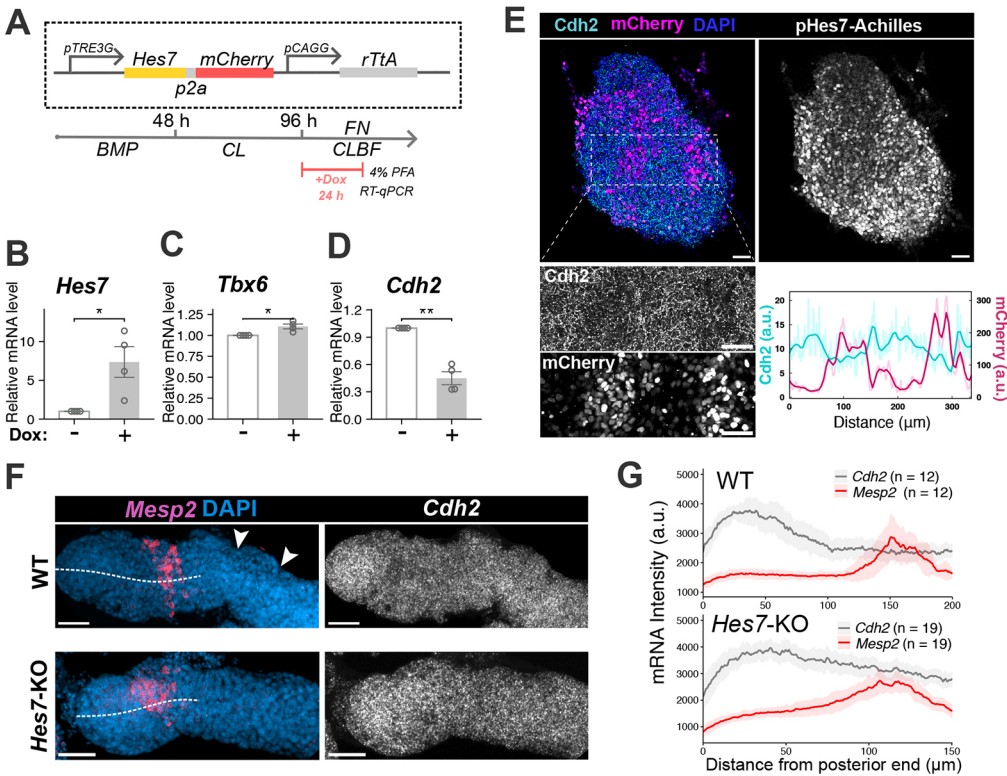

**Fig. 2. Hes7 represses *Cdh2* expression in anterior iPSM tissue.** (A) Scheme of the *Hes7*-overexpression vector (top) and the doxycycline (Dox) administration schedule (bottom). BMP, bone morphogenetic protein 4; FN, fibronectin; PFA, paraformaldehyde. (B-D) Expression of *Hes7* (B), *Tbx6* (C) and *Cdh2* (D) mRNA was quantified by real-time qPCR. Data are shown as the mean±s.e.m., *$P{\leq}0.05$, **$P{\leq}0.01$ (unpaired two-tailed *t*-test). *n*=4 independent experiments. (E) Representative immunofluorescence images of iPSM tissue overexpressing Hes7. The rectangular ROI is enlarged to visualize the distribution more clearly. Cdh2 protein intensity was plotted from left to right along with mCherry intensity. Bold lines represent smoothed intensity. Scale bars: 50 µm. (F) Representative HCR staining images of *Mesp2* mRNA expression (magenta), DAPI (blue) and *Cdh2* mRNA expression (grey). A dashed white line was drawn along the central axis of each iPSM, from the posterior end to the termination of the *Mesp2* stripe, to measure and plot intensity along the axis. Scale bars: 100 µm. Arrowheads indicate the segment boundaries. (G) Spatial plot of *Mesp2* mRNA expression along the posterior-anterior axis. Data are the mean±95% confidence interval, with the 2.5th and 97.5th percentiles shown for each data point. a.u., arbitrary units.

tailbud and newly formed somites while exhibiting diverse wave-like dynamic patterns in PSM samples (Fig. S5A). These patterns were coupled with the *Hes7* mRNA distribution to some extent; however, the differences between the 'on' and 'off' areas were much more subtle compared to the *Hes7* waves (Fig. S5A). Taken together, these observations indicate that *Cdh2* expression exhibits shallow waves that lead to a reduced level at anterior PSM, and this reduction requires periodical repression by Hes7.

Next, we evaluated the Cdh2 protein distribution in the PSM of WT mouse embryos to understand how it is involved during somitogenesis. E10.5 mouse PSM tissues were stained with antibodies for Cdh2 and Hes7, and we found that the majority of samples showed a stable gradient that was higher in the posterior region and lower in the anterior region (Fig. S5B). To confirm this Cdh2 protein expression pattern, we engineered the p*Hes7*-Achilles reporter ESC line with cDNA for a fluorescent protein, mScarlet, knocked-in at the C terminus of the *Cdh2* gene locus so that Cdh2-mScarlet fusion protein was expressed (Fig. 3A). In iPSM tissues induced from this ESC line, we were able to monitor Cdh2 protein levels and Hes7 oscillations in parallel. Live imaging recaptured the p*Hes7*-Achilles traveling waves and clear membrane localization of mScarlet signals (Fig. 3B), suggesting proper iPSM differentiation and successful integration of the fluorescent protein to the endogenous Cdh2 protein. As observed by

immunostaining, clear wave patterns were undetectable in time-lapse movies (Movie 3), which showed a relatively steady intensity in both posterior and anterior regions over time (Fig. 3C). Similarly, a gradient-like distribution along the posterior-anterior axis was maintained throughout the iPSM imaging (Fig. 3D, Movie 3) in which the protein level was high in the posterior PSM and decreased toward the anterior end (Fig. 3D). Such a difference between the mRNA and protein expression patterns could be explained by the protein stability. Hes7 protein exhibits a rapid turnover (about 22 min), which is key to its oscillatory expression (Hirata et al., 2004). Analysis using cycloheximide indicated a Cdh2 protein half-life of approximately 3 h in PSM cells (Fig. 3E-G), suggesting a much longer turnover time than Hes7. Taken together, these data support a model in which, although the mRNA exhibits dynamic expression patterns along the axis in PSM, the stability of the Cdh2 protein results in a slower turnover rate and a gradual reduction in its levels when cells progress to the anterior area after several pulses of Hes7 repression.

### High Cdh2 expression in PSM cells leads to prolonged Hes7 oscillations

We next examined whether Cdh2 expression levels impact Hes7 activity and oscillations in PSM cells. We generated cell lines carrying a Tet-ON system, in which Cdh2 expression is inducible by

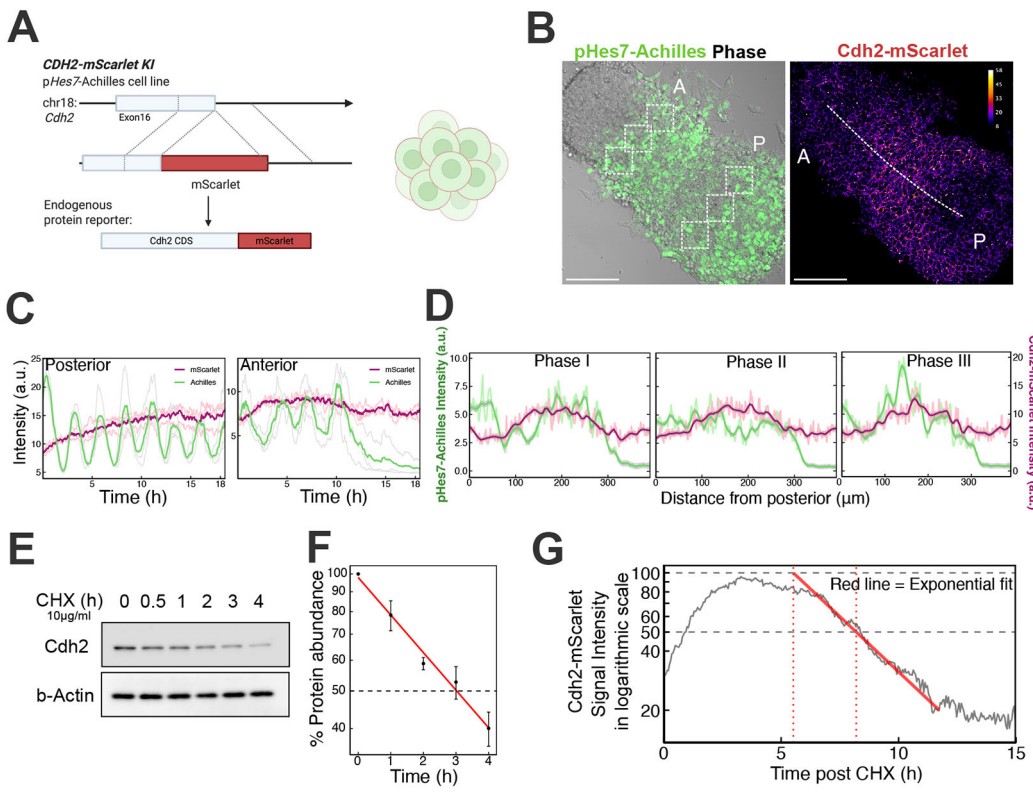

**Fig. 3. Cdh2 protein exhibits a relatively stable gradient along the posterior to anterior axis in iPSM.** (A) Scheme of the knock-in strategy for generating a Cdh2-mScarlet fusion protein reporter line. Created in BioRender by Jia, X. (2025). https://BioRender.com/f9n6ph3. This figure was sublicensed under CC BY 4.0 terms. (B) Representative snapshots of Achilles and mScarlet channels 6 h after starting live imaging (105-h post induction). Dashed squares indicate the ROIs for quantitative plot of intensity. A, anterior; P, posterior. Dashed line in the mScarlet channel is used for plotting intensity along the posterior-anterior axis. Scale bars: 100 μm. (C) Plots of the pHes7-Achilles intensity along time measured using the ROIs in B. The purple and green lines indicate the mean of intensities of Cdh2-mScarlet and pHes7-Achilles, respectively, from three ROIs, and the light pink and grey lines represent the intensity from each single ROI. (D) Spatial smoothed plots of intensity from pHes7-Achilles (green) and CDH2-mScarlet (purple) channels along the posterior-anterior axis. (E) CDH2 protein levels were quantified by western blotting of WT iPSM collected from a time series post-cycloheximide (CHX) administration. β-Actin was used as loading control. (F) Quantification of Cdh2 protein abundances shown as mean±s.e.m. for each time point and fitted to a logarithmic scale presented as the red line. *n*=3 independent experiments. (G) The Cdh2-mScarlet signal intensity in logarithmic scale measured by live imaging with CHX administration. The grey line is the mean value, and the red line is the exponential fitting to the slope of the mean intensity. Dashed grey lines denote the values (*y*=100, *y*=50) and corresponding time (*x*=5.52, *x*=8.2) is denoted by red dotted lines. a.u., arbitrary units.

the addition of doxycycline to the culture medium (Fig. 4A). Tet-ON-inducible overexpression of Cdh2 was confirmed by western blotting (see Fig. S7C). iPSM tissue overexpressing only mCherry that is lacking the Cdh2 coding sequence (Fig. S6A) exhibited no differences in morphology or elongation compared to the untreated samples (Fig. S6B). In contrast, iPSM tissue with Cdh2 overexpression lost its posterior-anterior axis upon doxycycline administration, with cells spreading more compared to the control group (Fig. 4B, Movie 4). The addition of doxycycline and exogenous protein overexpression led to weakened Hes7 reporter signals in both control and *Cdh2*-overexpressing iPSM tissue (Fig. S6G). However, mCherry expression alone did not significantly alter the Hes7 oscillation pattern, which was gradually weakened in the anterior region after several oscillation cycles (Fig. S6C-E,G, left), indicating that Hes7 expression stopped as the cells differentiated and exited from the PSM fate. However, in

*Cdh2*-overexpressing iPSM tissue, pHes7-Achilles activity was prolonged compared to the untreated iPSM and lasted until the end of the live imaging (Fig. 4C,D, Fig. S6F,G, right, Movie 4), suggesting that high Cdh2 expression may delay the timing of differentiation and exit from PSM property, maintaining Hes7 expression for a longer time. Indeed, the proportion of Tbx6-positive cells was increased by Cdh2 overexpression compared to WT samples while the Cdh2 KO samples showed a tendency of reduced Tbx6 proportion (Fig. 4E-G), indicating that the increased Cdh2 levels delayed PSM cell differentiation into somite-like tissues. Taken together, these results demonstrate that Cdh2 expression levels are crucial for maintaining PSM cell identity and regulating the timing of differentiation.

To investigate the mechanism by which Cdh2 regulates Hes7 oscillations, we next examined Notch signalling activity. In *Cdh2* KO iPSM tissue, expression of the Notch intracellular domain

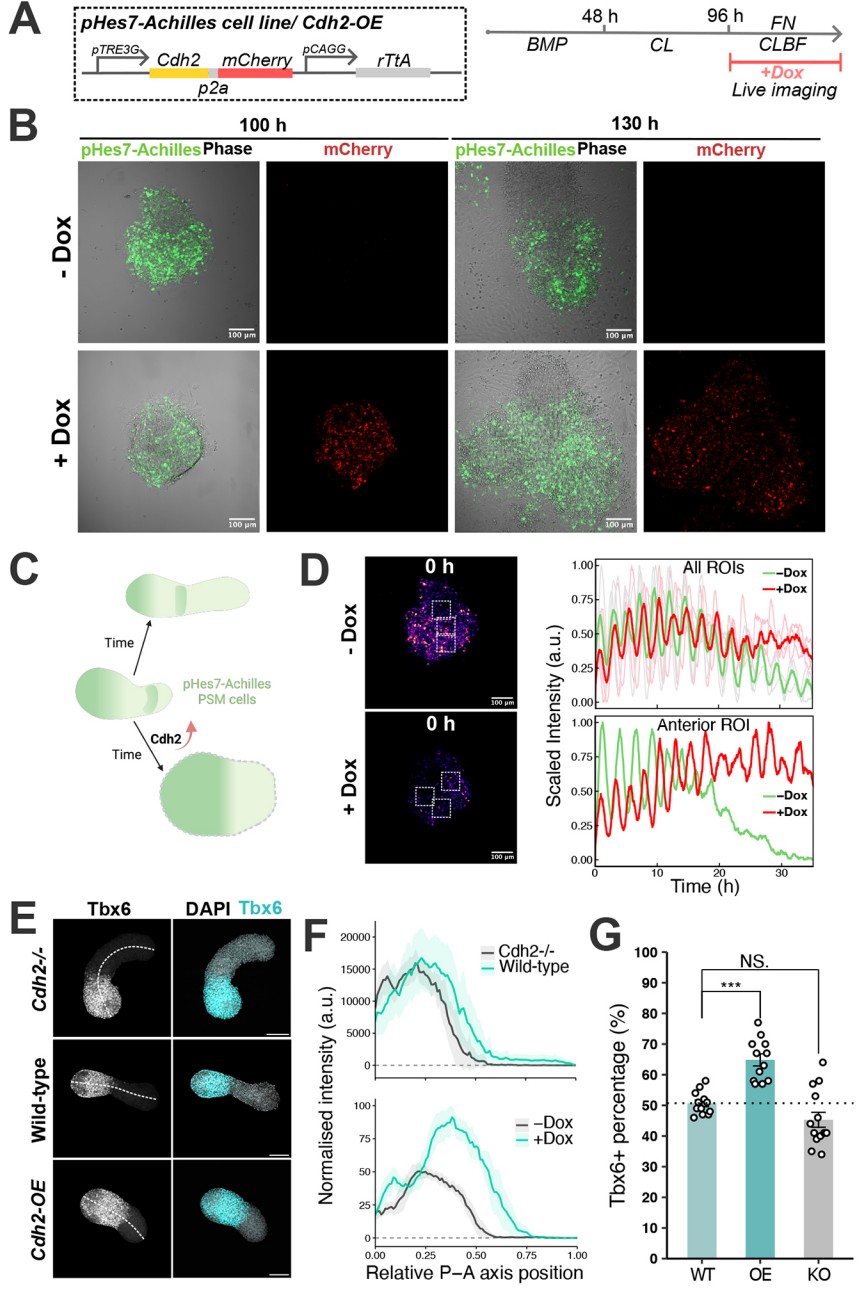

**Fig. 4. Delayed differentiation of iPSM tissue by *Cdh2* overexpression.** (A) Scheme of the overexpression (OE) vectors (left) and iPSM induction protocol (right). (B) Representative images of Achilles and mCherry channels at 100 h and 130 h with and without doxycycline (Dox) administration. (C) Scheme summarizing the phenotype of Cdh2-OE iPSM. Green shading represents Hes7 transcription activity. Created in BioRender by Jia, X. (2025). https://BioRender.com/dzb6pz7. This figure was sublicensed under CC BY 4.0 terms. (D) Representative images of pHes7-Achilles channels in Fire at 0 h after starting imaging (100-h post induction) (left) and the plot of pHes7-Achilles intensity along time measured using ROIs shown in the left panel (right top) and a single anterior ROI in Movie 4 (bottom). The purple and green lines indicate the mean of intensities from three ROIs, and the light pink and grey lines represent the intensity from each single ROI. *n*=3 ROIs. (E,F) Representative images of immunofluorescence staining (E) of iPSM tissue in CL medium at 108 h and corresponding signal measurement (dashed lines in E; data in F) along the posterior-anterior (P-A) axis of Tbx6, where 0 is the posterior end and 1 is the anterior end (*n*=12 iPSM tissues for each condition). Data are the mean±95% confidence interval, with the 2.5th and 97.5th percentiles shown for each data point. a.u., arbitrary units. (G) Bar chart comparing the Tbx6-positive percentage for each condition. The values are extracted from the plot shown in F; data are mean±s.e.m. \**P*≤0.05, \*\**P*≤0.01, \*\*\**P*≤0.001 (unpaired two-tailed *t*-test). NS, not significant. *n*=12 iPSM tissues for each condition.

(NICD) was not significantly affected (Fig. S7A,B), suggesting that Cdh2 is not essential for cell–cell contact-mediated Notch signalling. In contrast, in iPSM overexpressing *Cdh2*, NICD and Lfng levels were decreased, suggesting that Notch signalling could be weakened by Cdh2 overexpression (Fig. S7C-E). However, because Notch signalling is required for Hes7 oscillations, weakened activity of this pathway cannot explain why Hes7 oscillations were prolonged by *Cdh2* overexpression. These results suggest that a different pathway from Notch signalling may be involved in Cdh2-controlled Hes7 oscillations.

## FGF signalling activity is related to Cdh2 levels in PSM cells

The posterior-anterior gradient of FGF signalling and the oscillatory activity of its downstream target, ppERK, are crucial for the precise positioning of segmental boundaries (Dubrulle and Pourquié, 2004; Niwa et al., 2007; Wahl et al., 2007; Anderson et al., 2020). In addition, inhibition of FGF signalling can completely abolish Hes7 expression in the PSM (Niwa et al., 2007). Previous studies in other cell types have shown that Cdh2 can modulate FGF activity by interacting with FGFR1 (Fig. 5A), which controls cell proliferation, differentiation and migration (Williams et al., 2001; Suyama et al., 2002; Sanchez-Heras et al., 2006; Takehara et al., 2015; Kon et al., 2019). However, the relationship between Cdh2 levels, the FGF gradient, and ppERK oscillations in PSM cells remains to be elucidated.

To determine whether the early arrest of Hes7 oscillations in *Cdh2* KO iPSM tissue was associated with disrupted FGF signalling, ppERK levels were quantitatively assessed by western blotting. We prepared protein extracts from pooled 96-h iPSM (Fig. 5A, bottom),

in which most cells exhibited PSM properties. The ppERK/ERK ratio was reduced in *Cdh2* KO iPSM tissue compared to WT tissue (Fig. 5B,D). In addition, the levels of phosphorylated FGFR1 (pFGFR1), an active form of FGFR1, were also reduced in *Cdh2* KO iPSM tissue compared to WT tissue (Fig. 5B,C), indicating that the absence of Cdh2 leads to downregulation of FGF activity.

We next examined spatial patterns of ppERK activity along the posterior-anterior axis. ppERK distribution was analysed in iPSM tissue at 108 h post-induction (Fig. 5E,F). Immunostaining analysis revealed high ppERK levels in the posterior region of WT iPSM tissue, followed by a sharp decrease at the anterior PSM region and an increase in the differentiated half, including somite-like structures (Fig. 5E, WT). In contrast, *Cdh2* KO iPSM tissue exhibited lower average ppERK levels compared to WT cells (Fig. 5E). These results suggest that ERK signalling activity was attenuated in the absence of Cdh2. Conversely, when *Cdh2* was overexpressed, overall ppERK levels were increased throughout the iPSM compared to the control, reducing the posterior-anterior gradient of ppERK levels (Fig. 5F). These results suggest that Cdh2 plays an important role in FGF signalling activity in the PSM.

To investigate whether physical interaction between FGFR1 and Cdh2 is required for FGF signalling, we sought to inhibit this interaction by a blocking peptide assay, in which a peptide containing the identified sequence on Cdh2 protein responsible for interacting with FGFR1 (Williams et al., 2001) was added to the iPSM culture (Fig. S8A-F). Addition of a blocking peptide reduced the ppERK levels (Fig. S8E,F) and Tbx6 proportion (Fig. S8B,F) without affecting iPSM morphology (Fig. S8D). We also generated a mutant,

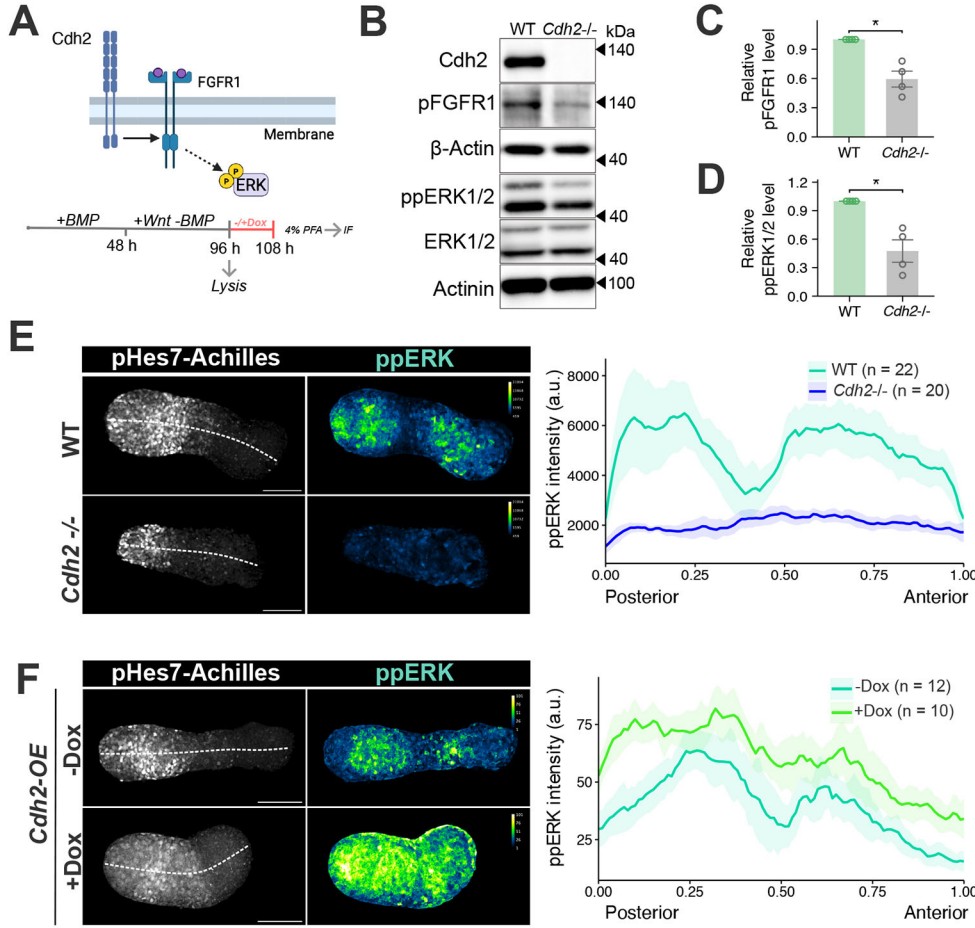

**Fig. 5. FGF-ppERK activity is related to Cdh2 expression levels.** (A) Illustration of the relationship between Cdh2 and FGF signalling (top) and protocol for measuring ppERK in iPSM tissue by western blotting and immunofluorescence staining (bottom). Created in BioRender by Jia, X. (2025). https://BioRender.com/k592awn. This figure was sublicensed under CC BY 4.0 terms. (B-D) ppERK levels were quantified by western blotting. WT and *Cdh2*$^{-/-}$ iPSM samples (*n*=4 independent experiments) were collected at 96 h. Data are shown as mean±s.e.m. *$P \leq 0.05$ (unpaired two-tailed *t*-test). (E) Representative immunofluorescence images (left) and plots (right) comparing ppERK spatial profiles between WT control and *Cdh2*$^{-/-}$ iPSM tissues, plotted on a normalized major axis where 0 is the posterior end and 1 is the anterior end. Dashed lines indicate the axis from which the measurements were taken. Scale bars: 100 µm. (F) Representative immunofluorescence images (left) and plots (right) comparing ppERK spatial profiles between control [−doxycycline (Dox)] and *Cdh2*-overexpressing (+doxycycline) iPSM tissues. Scale bars: 100 µm. a.u., arbitrary units. In E and F, shaded areas represent the 2.5th and 97.5th percentiles and solid lines represent the mean intensity.

in which extracellular domain 4 (ECD4), required for the interaction with FGFR1, was removed from Cdh2 (Fig. S8G). This deletion also reduced the ppERK levels (Fig. S8H,J) and Tbx6 proportion (Fig. S8H,K) without affecting the iPSM morphology (Fig. S8I). These results suggest that the direct physical interaction between Cdh2 and FGFR1 is responsible for maintaining downstream ppERK activity and PSM cell properties. These findings together indicate that Cdh2 levels are closely tied to FGF/ppERK activity, which subsequently influences the differentiation of PSM cells.

## Cdh2 expression influences *Mesp2* patterning and PSM cell maturation

Along the posterior-anterior axis, PSM cells transition from an immature state, gradually differentiating through intermediate regions and ultimately committing to somite fate at the anterior-most end. FGF signalling plays a crucial role in this process, with high activity in the posterior region to maintain PSM cells in an immature state and restricted activity in the anterior region to enable differentiation. In *Cdh2* KO iPSM tissue, the early cessation of Hes7 oscillations and the repressed ERK activity suggest that, in the absence of *Cdh2*, PSM cells prematurely adopt a more anterior-like property.

To test this hypothesis, we analysed the periods of Hes7 oscillations in WT and *Cdh2* KO iPSM tissues (Fig. 6A,B). In WT iPSM tissue, the oscillation periods in the anterior regions were progressively longer than those in the posterior regions (Fig. 6B). Notably, the periods in the posterior regions of *Cdh2* KO iPSM tissue were longer than those observed in posterior WT iPSM tissue, but similar to the WT anterior oscillations (Fig. 6B). To determine whether these slowed oscillations arise from intrinsic alterations or tissue-level factors, oscillation periods were measured in single iPSM cells by dissociation culture (Fig. 6C). Single-cell analysis of the peak-to-peak periods from individual cell tracks revealed that pHes7-Achilles reporter oscillations in *Cdh2* KO cells are slower than those in WT cells (Fig. 6D), supporting the conclusion that the loss of Cdh2 confers a more anterior-like property (slower oscillations) on iPSM tissue.

To investigate further PSM cell maturation and differentiation into somites, *Mesp2* expression patterns were compared between WT, *Cdh2* KO and *Cdh2*-overexpressing iPSM tissues. *Mesp2* is expressed in the prospective somite regions in the anterior PSM, where Notch activity and clock oscillations cease. Since FGF signalling represses *Mesp2* expression (Oginuma et al., 2008), we

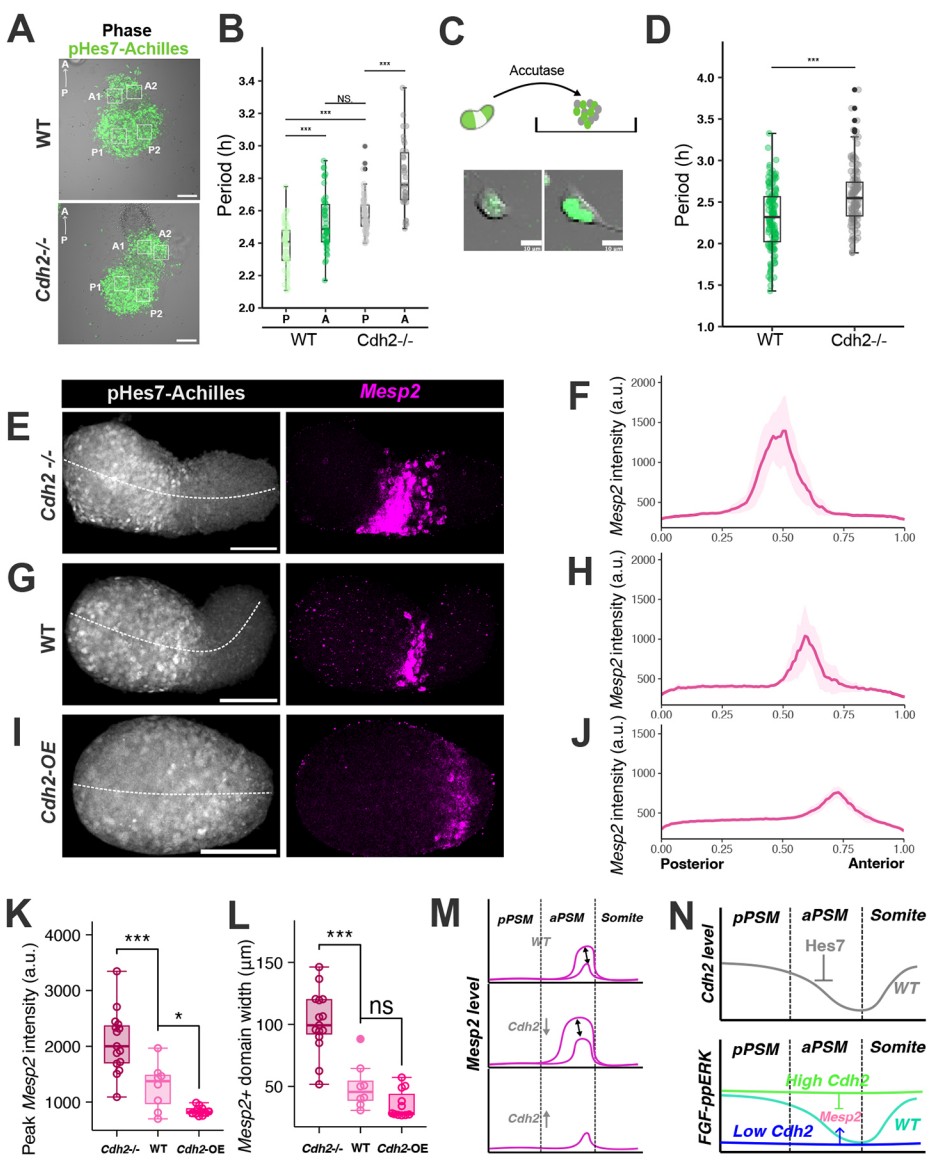

**Fig. 6. More anterior properties in *Cdh2* KO iPSM tissue than WT.** (A,B) Oscillation periods of fluorescence of the pHes7-Achilles reporter in iPSM tissue. The periods were exacted from the ROI (A) plot by calculating the peak-to-peak intervals. A, ROI at the anterior iPSM; P, ROI at the posterior iPSM. ***P≤0.001 (ANOVA). NS, not significant. n=3 independent experiments. Scale bars: 50 µm. (C) iPSM tissues were cultured following the iPSM protocol. At 96 h, iPSM tissue was dissociated into single cells using Accutase and plated on fibronectin-coated plates. Scale bars: 10 µm. (D) Oscillation periods of fluorescence of the pHes7-Achilles reporter in dissociated iPSM cells. ***P≤0.001 (unpaired two-tailed *t*-test). n=3 independent experiments. (E,G,I) Representative HCR staining images of *Mesp2* mRNA expression (magenta) and pHes7-Achilles activity (grey). Scale bars: 100 µm. (F,H,J) Spatial plots of *Mesp2* mRNA expression along the posterior-anterior axis (dashed lines in E,G,I) for Cdh2 KO (F), WT (H) and *Cdh2*-overexpressing (J) iPSM. Data are the mean±95% confidence interval, with the 2.5th and 97.5th percentiles shown for each data point. n=17 iPSM for Cdh2−/−; n=10 iPSM for WT; n=12 iPSM for Cdh2-OE. (K) Maximal signal intensity of *Mesp2* mRNA expression extracted from the plots shown in F,H,J. (L) The spatial length of *Mesp2* mRNA expression in iPSM tissue calculated from the plots shown in F,H,J. For K,L, *P≤0.05, ***P≤0.001 (unpaired two-tailed *t*-test). ns, not significant. (M) Scheme summarizing the phenotypes of *Mesp2* expression. The magenta lines represent the expression intensity along the axis, and the black double-headed arrows between the magenta lines indicate the fluctuations between samples. (N) Summary of perturbations of *Cdh2* expression in iPSM tissue and the resultant phenotypes. a.u., arbitrary units.

hypothesized that *Cdh2* disruption would alter *Mesp2* expression. WT, *Cdh2* KO and *Cdh2*-overexpressing iPSM tissues were induced in parallel, kept in CL medium to allow differentiation (without switching to CLBF), and fixed at 108 h post-induction. *Mesp2* mRNA was detectable in all conditions (Fig. 6E,G,I) and the *Mesp2* expression pattern was observed as a stripe in each iPSM, corresponding to positions where the termination of pHes7-Achilles reporter signal occurs. The distribution of *Mesp2* along the posterior-to-anterior axis was quantified by measuring intensity along a midline drawn through the centre of each iPSM (Fig. 6E,G,I, dotted lines). Compared to WT tissue, the *Mesp2* stripe in *Cdh2* KO iPSM tissue was shifted posteriorly, whereas the stripe in *Cdh2*-overexpressing iPSM tissue was shifted anteriorly (Fig. 6E-J). This suggests accelerated differentiation in the absence of *Cdh2* and delayed differentiation with high *Cdh2* expression.

To characterize further the dynamics of *Mesp2* expression, peak amplitude (Fig. 6K) and stripe width of expression (Fig. 6L) were analysed. The *Mesp2* stripe length was quantified by measuring the distance between the defined start and end points of expression intensity. In *Cdh2* KO iPSM tissue, peak amplitude and stripe width were increased relative to WT tissue (Fig. 6E-H,K,L). Conversely, in *Cdh2*-overexpressing iPSM tissue, *Mesp2* expression amplitude was reduced, and the fluctuations observed in WT and KO iPSM tissues were absent (Fig. 6I-L). We also found that *Mesp2* expression intensity and width increased significantly in *Cdh2* KO compared to WT tissue (Fig. 6K,L), suggesting that *Mesp2* may continue to oscillate in the anterior region of *Cdh2* KO iPSM tissue as in WT iPSM tissue (Fig. 6M). It is likely that the posterior shift and expansion of *Mesp2$^+$* region in the absence of *Cdh2* was due to reduced ppERK activity and premature differentiation of PSM cells. This observation was further supported by staining for the somite markers *Uncx4.1* (*Uncx*) and *Meox1*, which were also detected in the PSM region when *Cdh2* was absent (Fig. S9B,D), but not in WT tissue (Fig. S9A,C). In contrast, in tissue with elevated *Cdh2* expression, *Mesp2* expression was constrained to low levels and shifted anteriorly (Fig. 6I,J,M), indicating that low Cdh2 levels at the anterior end are essential for PSM cells to differentiate into somite cells (Fig. 6N).

## DISCUSSION
In this study, we found that Hes7 periodically represses *Cdh2* transcription in the PSM cells. Given the longer half-life of Cdh2 protein than that of Hes7, this repression would fine-tune Cdh2 protein levels in the anterior PSM, regulating FGF signalling activity and *Mesp2* expression and facilitating cell differentiation (Fig. 6N). Cdh2 expression is then restored in newly formed somites to establish their structural integrity quickly.

The dynamic expression of *Hes7* plays essential roles in somitogenesis, thereby it is an important factor in understanding how cells govern precise temporal and spatial information. Evidence linking *Hes7* to major signalling pathways, e.g. Notch, FGF and Wnt, has emerged over recent decades, providing insights into how Hes7 interacts with other clock and gradient systems. However, data remain limited within the known segmentation clock framework. Here, we conducted ChIP-seq on mouse PSM cells using an anti-Hes7 antibody, revealing a comprehensive view of the *Hes7* regulatory landscape. The large cohort of genes identified not only included known *Hes7* targets but also revealed numerous genes previously unrelated to the segmentation clock, raising the possibility that Hes7 regulates a wider variety of genes in the PSM than previously thought. Interestingly, direct binding of Hes7 to the *Dll1* locus was undetectable, suggesting that the signal-sending

process through the Delta-Notch pathway is indirectly regulated by Hes7. This is in contrast to the gene regulatory network of the zebrafish segmentation clock, in which core oscillator *her1/7* proteins directly regulate transcription of *deltaC* (Oates et al., 2012). One possibility linking Hes7 to Delta-Notch is an indirect regulation through Lfng, which is consistent with the observation of a ChIP-seq peak at the *Lfng* locus and the previous report describing direct Hes7 binding to the *Lfng* promoter (Bessho et al., 2003).

By screening a subset of the Hes7 target genes, we identified *Cdh2* as a potential regulator of Hes7 oscillations. While *Cdh2* has been implicated in somite structure formation and cell movement, its role in oscillatory gene expression has not been explored. Our investigation demonstrates that *Hes7* plays a crucial role in modulating *Cdh2* expression in PSM cells, which in turn influences FGF signalling and PSM cell maturation. These findings align with the differentiation potential of cadherins observed recently (Mayran et al., 2023 preprint) and suggest that Hes7-controlled *Cdh2* expression levels modulate the timing and patterning of somitogenesis, revealing a previously unrecognized layer of regulation within the segmentation clock.

Previous studies have shown that FGF signalling exhibits two patterns: ligand expression displays a spatial gradient from the posterior to anterior PSM (Dubrulle et al., 2001; Dubrulle and Pourquié, 2004), while FGF activation (ppERK) cycles in an on-off manner in the PSM (Niwa et al., 2007; Wahl et al., 2007; Niwa et al., 2011; Simsek et al., 2023). The cyclical oscillations of FGF activation disappear in *Hes7*-null PSM cells, leading to irregular somite formation in mice (Niwa et al., 2011). Similar phenotypes have been observed in *her1/her7*-null zebrafish PSM cells, where proper boundary formation can be restored by periodic inhibition of FGFR/ERK activity (Simsek et al., 2023). *Dusp4*, an inhibitor of ppERK activity, is a downstream target of *Hes7* (Fig. S2) and oscillates in response to *Hes7* activity (Niwa et al., 2007; Niwa et al., 2011). However, possibly due to compensatory mechanisms and the presence of other regulators, studies on *Dusp4*-deficient mice have not yet clarified how *Hes7* regulates FGF activation (Auger-Messier et al., 2013). In our ChIP-seq analysis, *Cdh2* was identified as a target of Hes7, and our findings suggest that *Cdh2* may modulate ppERK signalling dynamics in the PSM. Evidence from studies in other cell types suggests that Cdh2 protein interacts with the extracellular domain of FGFR1 on the cell membrane (Williams et al., 2001; Sanchez-Heras et al., 2006), tightly correlating with downstream ppERK activation, indicating that *Hes7* can indirectly fine-tune FGF activity by controlling *Cdh2* expression. This interaction positions *Cdh2* as a key mediator that connects signalling gradients with the oscillatory clock.

Here, we revealed that expression patterns of *Cdh2* mRNA are temporally dynamic and those of Cdh2 protein are spatially graded. Wave-like distribution of *Cdh2* mRNA levels within the PSM follows *Hes7* mRNA expression, as observed in HCR analysis of E9.5 mouse embryos. The relatively low expression of *Cdh2* in the PSM, compared to surrounding tissues, such as the tailbud, notochord and neural tube, may explain why previous transcriptomic studies failed to detect its dynamic expression. However, Cdh2 protein displays a rather stable gradient pattern that is high in the posterior region and low in the anterior region, probably due to its stability, and this gradient seems to be important for proper FGF signalling as *Cdh2* upregulation expands the Hes7 oscillation region anteriorly. Thus, our findings uncover a more complex network than previously appreciated involving the adhesion molecule Cdh2 in somitogenesis.

Although there is no direct evidence for a physical interaction between Cdh2 and Notch1, Cdh2 is known to recruit and be cleaved

by presenilin 1 (Marambaud et al., 2003), which mediates the ligand-induced cleavage of Notch receptors. In the present study, we observed no significant change in NICD levels in *Cdh2* KO iPSM tissue, while NICD and Lfng levels were slightly downregulated in *Cdh2*-overexpressing iPSM tissue (Fig. S7). This rules out Notch as the cause of *Hes7* expression cessation. However, altering *Cdh2* levels may disrupt the balance of Notch receptor cleavage, potentially contributing to changes in *Mesp2* patterning, as NICD also regulates *Mesp2* expression (Morimoto et al., 2005; Oginuma et al., 2008). The reduction of NICD via *Cdh2* overexpression could decrease *Lfng* expression, leading to desynchronization, as *Lfng*, a well-known clock gene downstream of *Hes7*-induced repression and Notch-induced activation, is required for the maintenance of synchronized oscillations in PSM cells (Okubo et al., 2012; Yoshioka-Kobayashi et al., 2020).

Besides the effects on signalling pathways, intriguingly, we observed that reduced Cdh2 levels resulting from Hes7 repression in iPSM cells (Fig. 2C,E) led to clustering of these populations, indicating that imbalanced adhesion triggered cell sorting. Moreover, the iPSM lost a clear posterior-anterior spatial axis upon excessive Cdh2 expression (Fig. 4B, Fig. S6F). These findings suggest that a homeostatic level of adhesion strength along the axis is required in the PSM and may be regulated by the clock gene *Hes7*, potentially enabling neighbouring cells to migrate and differentiate in a synchronized manner. Further analyses are needed to dissect these mechanisms in greater detail.

To advance the dissection of clock gene function *in vitro*, we aimed to establish an *in vitro* system that enables more targeted manipulation of clock gene function. The iPSM system with a pHes7 reporter was previously used for drug screening to identify compounds that modulate the segmentation clock (Matsumiya et al., 2018; Yoshioka-Kobayashi et al., 2020). However, the complexity of the underlying signalling networks limits its utility in pinpointing specific regulatory components. In the present study, we combined this system with gene knockout and overexpression approaches, providing a powerful method for more precise and direct exploration of candidate genes involved in the segmentation clock. This system is also compatible with live imaging at the single-cell level, which can offer temporal and spatial information to better understand complex phenotypes related to synchronized oscillations. Taken together, our approach provides a versatile platform for dissecting the molecular mechanisms of the segmentation clock and may facilitate the discovery of further regulators involved in vertebrate somitogenesis.

## Study limitations
We revealed that Hes7 represses Cdh2 expression in the PSM and that the loss of Cdh2 would affect the differentiation front. However, the direct relationship between Hes7 binding to the *Cdh2* locus and segmentation clock function has not yet been addressed, as we have not specifically removed the binding site.

We also found that Cdh2 supports FGF signalling in the PSM, but the precise mechanism remains to be clarified. A physical interaction between Cdh2 and FGFR1 could stabilize the active form of FGFR1 (pFGFR1), consistent with our observation that pFGFR1 was decreased in Cdh2 KO iPSM tissue.

In addition, altering cell–cell adhesion may influence intracellular mechanical properties, which have also been implicated as important factors in the segmentation clock (Hubaud et al., 2017). Although we cannot currently rule out this possibility, future studies will be needed to clarify whether Cdh2 modulation affects such mechanical properties.

## MATERIALS AND METHODS
### ChIP-seq of E10.5 mouse PSM
Intact PSMs with one or two somites were collected from E10.5 mouse embryos and submitted to the Active Motif custom service for ChIP-seq analysis (small scale). This was carried out with 30 µg of PSM cell chromatin and anti-Hes7 antibody (Bessho et al., 2003). The ChIP DNAs were prepared as standard Illumina ChIP-Seq libraries and sequenced to generate >5 million reads; 10.5 million reads were aligned to the mouse genome (mm10); after removal of duplicate and non-uniquely mapped reads, ~4.1 million alignments were obtained. A signal map capturing fragment densities along the genome was generated and visualized in the Integrated Genome Browser (IGB). MACS peak finding was performed to identify the most significant peaks. Using their default cutoff of $P$-value $1e^{-7}$, 913 peaks were identified for Hes7 (indicated by black lines in Fig. S1A). Visualization and annotation of the peaks were carried out using the IGV online tool (https://igv.org/app/) and the ChIPseeker R package (Wang et al., 2022; Yu et al., 2015). All animal experiments were approved by the Institutional Animal Care and Use Committee at RIKEN.

### Plasmid construction
Standard molecular biology techniques were used in this study. All DNA constructs mentioned in this study are based on the Tol2 transposon vector system, a gift from the Kawakami lab (Kawakami, 2007; Yagita et al., 2010). For the knockout screening of candidate genes, the mutations were introduced into the mouse ESC genome using the CRISPR-Cas9-based genetic editing system containing the high-fidelity nuclease eSpCas9(1.1) (Addgene plasmid #71814) (Ran et al., 2013; Slaymaker et al., 2016). We constructed a plasmid in which eSpCas9(1.1)-P2A-Puro is driven by a pEFS promoter and sgRNA expression is driven by a glutamine-tRNA promoter (Mefferd et al., 2015). The sgRNA sequence was inserted into the vector through a BsmBI restriction enzyme site.

The selected sgRNA sequences (Table S2) for targeting the genes of interest were generated and filtered by a pipeline using online prediction tools. First, a list of the sequences were generated by inDelphi (Shen et al., 2018) after choosing the preferred editing positions, editing precision and frameshift frequency. Then, the options with high off-targets were excluded by referring to CRISPOR (Concordet and Haeussler, 2018). Lastly, the sgRNA sequence was determined with the highest activity score predicted by DeepSpCas9 (Kim et al., 2019) for the combination of eSpCas9(1.1) and glutamine-tRNA promoter. The negative control sgRNA sequences were taken from the study (Joung et al., 2017).

For generating the Cdh2-mScarlet knock-in cell lines, we engineered the donor vector for knock-in of the fluorescent protein mScarlet into the C-terminal portion of the *Cdh2* coding sequence by inserting the donor sequence into the Bluescript II SK(−) plasmid. The donor sequence contained the mScarlet cDNA sequence flanked by 1000-bp upstream (5′ homology arm) and 1000-bp downstream (3′ homology arm) the stop codon of the Cdh2 C terminus. For selection of transfected cells, a sequence of pEFS-iPFP-p2a-Neomycin was inserted after the 3′ homology arm described above.

The overexpression vectors were constructed based on a Tet-ON system containing an expression cassette of rtTA_3G-P2A-Puro under the promoter pCAGG and the expression cassette for NLS-mCherry driven by the TRE_3G promoter. HA-Cdh2(CDS)-P2A or HA-Hes7(CDS)-P2A was inserted in front of the NLS-mCherry sequence to generate the plasmids for overexpressing Hes7 or Cdh2 in mouse ESCs (mESCs). The empty construction was used to generate cell lines as a negative control.

### mESC culture
All mESCs used in this study were derived from the E14tg2a cell line (RIKEN Bio Resource Center #RBRC-AES0135). We maintained mESCs in an a2i condition (Choi et al., 2017; Yagi et al., 2017) in which NDiff227 medium (Takara Bio, Y40002) was supplemented with 3 µM CHIR99021 (Sigma-Aldrich, SML1046), 1.5 µM CGP77675 (Cayman Chemical, 21089), 1000 U/ml leukaemia inhibitory factor (StemSure LIF; FUJIFILM Wako 199-16051), 100 U/ml penicillin and 100 µg/ml streptomycin (Nacalai Tesque, 09367-34) in a humidified incubator at 37°C and 5% $CO_2$. The cells were passaged every 2 days for routine maintenance.

## Cell line generation

The plasmids were transfected into the reporter cell line (Isomura et al., 2025) following the protocol described in previous studies (Isomura et al., 2025; Maeda et al., 2023). A 100 μl mixture of 0.5 μg designed plasmids, 0.5 μg pCAGGS-mT2TP plasmids and 3 μl ViaFect Transfection Reagent (Promega, E4981) diluted in Opti-MEM™ (Gibco, 31985070) was added to a 100 μl $2 \times 10^5$ cell solution and plated onto a 12-well plate. For selecting successful integrated cells, the supernatant was removed and replaced with fresh a2i medium supplemented with 1 μg/ml puromycin after 2 days. The population of transfected cells was further passaged and plated sparsely to prepare for single-cell cloning and genotyping. At least three independent cell lines were generated for each gene.

For generating Cdh2-mScarlet knock-in cell lines, transfected cells were first incubated with 200 μg/ml neomycin in culture medium and then sorted by iRFP. Then, the sorted cells were subjected to single-cell cloning, and successful knock-in cell lines were confirmed by genotyping.

For generating mutant cell lines with Cdh2 ECD4 (a.a.515-604) removal (Kon et al., 2019), pHes7-Achillles mESCs were transfected with the sgRNA-expressing plasmids together with a repair fragment (TCCTTATTTTGCCCCAAATCCTAAAATCATTCGCCAAGAGGAAG-GCCTCCACGCAGGTACCGTGTTACCTCAAGAGGCGGAGACCTG-TGAAACTCCAGAACCCAACTCAATTAACATCACA). Then, the cells were sorted by GFP and subjected to single-cell cloning and genotyping for confirmation of the deletion.

## iPSM

Organoids with a PSM property were induced following a protocol described previously (Isomura et al., 2025; Matsumiya et al., 2018). Following the mESC-maintaining procedures, instead of a2i medium, cells were resuspended in NDiff227 medium supplied with 20 ng/ml BMP4 (R&D Systems, 314-BP-010/CF). Then, $2.4 \times 10^4$ cells were plated in each well of a 96-microwell plate (EZSPHERE 96-Well Plate; IWAKI, 4860-900) (Sato et al., 2016) to give about 80 aggregates (each comprising about 300 cells) in each well. After 48 h, the aggregates were washed with wash medium (DMEM/F12 medium; Nacalai Tesque, 0517715) supplemented with 0.1% bovine serum albumin (Thermo Fisher Scientific, 15260037) and transferred into CL medium for a further 48-h incubation. To make CL medium, DMEM (Nacalai Tesque, 08489-45) basal medium was supplemented with 1× GlutaMax, 1 mM sodium pyruvate (Thermo Fisher Scientific, 11360-070), 1 mM nonessential amino acids (Thermo Fisher Scientific, 11140-050), 1× B-27 without vitamin A, 100 U/ml penicillin, 100 μg/ml streptomycin, 1 μM CHIR99021, 100 nM LDN193189 (Sigma-Aldrich, SML0559), and a 0.75% final concentration of DMSO (Sigma-Aldrich, 472301-100ML). For immunostaining and HCR to check differentiation, the iPSM samples were kept in CL medium for the required time prior to fixation. For live imaging of oscillations, the iPSM samples were transferred to the CLBF medium, which was supplemented further with small inhibitor molecules [2.5 μM retinoic acid signalling inhibitor BMS493 (Sigma-Aldrich, B6688-5MG) and 50 ng/ml HS basic FGF (Thermo Fisher Scientific, PHG0369)] to keep iPSM cells in the posterior state for prolonged periods. To observe somitogenesis, iPSM samples were embedded in 10% Matrigel (Corning, 356231)/NDifff277 solution and placed on a Lipidure (Amsbio, CM5206)-coated plate. The plate was kept inside a humid incubator before fixation without disturbance.

## Time-lapse live imaging and image processing

To capture dynamics of pHes7 activity, iPSM samples were resuspended in CLBF medium and placed homogeneously in each well of a μ-Dish 35-mm Quad dishes (ibidi, 80416) previously coated with 10 μg/ml fibronectin (Corning, 354008) solution for more than 1 h in an incubator and washed with PBS before transferring the cells. To ensure cell attachment, the dish was placed in a humid incubator for 3 h prior to imaging. To control for technical effects during iPSM induction and imaging, control cells were always prepared and handled in the same conditions. For KO screening, three cell lines for each mutant were always induced and imaged in the same condition as the WT cells. For the overexpression experiments, iPSM samples carrying the construct were induced and separated into two groups and doxycycline was added to one of them. In preparation for the imaging of

single, dissociated iPSM cells, iPSM organoids were collected and incubated in Accutase for 7 min in a humid incubator. Then, the cells were dissociated by gently pipetting and recollected by centrifugation (217 *g* for 5 min). The cells were resuspended in CLBF medium, and a low number of cells were seeded in each well of a μ-Slide 18-well glass-bottom chamber (ibidi, 81817) pre-coated with 5 μg/ml fibronectin solution. The chamber was directly brought to a microscope for imaging.

Live-cell imaging of the pHes7-Achilles reporter in iPSM organoids was performed using an inverted fluorescence confocal microscope (Zeiss LSM980) equipped with a humid incubator with 5% $CO_2$ and at 37°C. The movies in this study were all taken using a 20× dry objective lens. Achilles was excited with a 514-nm Argon laser and captured every 3 min. The signals from the selected regions of interest (ROIs) were plotted and analysed using Fiji/ImageJ. To analyse the periods in single cells, cells from brightfield images were tracked manually using TrackMate (Tinevez et al., 2017) in Fiji/ImageJ. Hilbert transform was performed to obtain an instantaneous oscillation phase. Periods were quantified by peak detection on detrended and smoothed intensity. For each ESC line, live-cell imaging and analyses were performed at least three times.

## Inhibition of protein synthesis by cycloheximide

The half-life of Cdh2 protein was evaluated in iPSM cells by a cycloheximide assay. Cycloheximide was added to iPSM in CL medium starting from 96 h (final concentration 10 μg/ml) for 0, 0.5 h, 1 h, 2 h, 3 h and 4 h. All samples were collected and lysed for analysis by western blot using anti-Cdh2 and anti-actin antibodies.

## Peptide synthesis and purity

A synthetic peptide was prepared and obtained from the Peptide Synthesis Service at the Support Unit for Bio-Material Analysis in RIKEN Center for Brain Science, Research Resources Division (RRD). The sequence (Ac-WLKIDPVNGQI-NH2) was submitted to the Support Unit and the peptide sample was returned as lyophilized powder with >90% purity. For blocking experiments, the peptide was dissolved in DMSO at 200 mg/ml and aliquoted into small volume stocks kept at −80°C. Since the peptide has a low hydrophilicity, dilution in medium (200 μg/ml) was prepared a few hours prior to the experiments, vortexed and transferred to a culture dish for incubation inside a 37°C humid incubator. The same volume of DMSO was added to the neighbouring well as a control.

## Western blotting and qPCR analysis

Western blotting was performed as previously described (Kobayashi et al., 2019). The amount of each protein band relative to that of an actin or actinin band was quantified. The following primary antibodies were used for western blotting: rabbit anti-actin (Sigma-Aldrich, A2066), rabbit anti-pFGFR1 (Cell Signaling Technology, 52928), rabbit anti-Erk1/2 (Cell Signaling Technology, 9102), rabbit phospho-p44/42 MAPK (Erk1/2) antibody (Cell Signaling Technology, 9101), rabbit anti-Cdh2 (Cell Signaling Technology, 13116) and rabbit anti-a-actinin (Cell Signaling Technology, 6487). Secondary antibodies were HRP-conjugated anti-rabbit antibodies (GE Healthcare, NA9340-1ML). For detecting phosphorylated target protein, the handling for specific antibodies was adjusted by referring to the Cell Signaling Technology primary antibody product webpage. Compared to the general protocol, TBST was used instead of PBST. After blocking with 5% skim milk/TBST solution, the membranes were washed three times for 5 min each with TBST. 5% BSA/TBST solution is used instead of skim milk for the primary antibody dilution. For qPCR, total RNA was extracted using the RNeasy Plus mini kit (QIAGEN) and analysed on a QuantStudio 12K system (Applied Biosystems). RNA levels were normalized against the corresponding levels of β-actin and *Gapdh* mRNA. Primers (Table S2) for real-time PCR were purchased from Eurofins.

## Sample fixation for staining

For iPSM organoids grown in 10% Matrigel, the medium was not removed and 1 ml 4% paraformaldehyde in PBS was added directly to 300 μl Matrigel solution for fixation overnight at 4°C. For iPSM organoids grown in CL medium or CLBF medium on a fibronectin-coated dish, the medium was removed and washed once with PBS before fixation with 4%

paraformaldehyde in PBS overnight at 4°C. After fixation, all samples were washed three times for 5 min each wash in PBST [PBS with 0.1% Tween 20 (v/v)], transferred to 100% methanol and incubated at −30°C overnight or stored for longer. Before proceeding to the next step, the samples were removed from methanol and washed three times in PBST at room temperature. For mouse embryos, the handling was carried out following a previous protocol (Alanentalo et al., 2007). To quench the autofluorescence from blood cells and tissues, embryos were subjected to methanol:DMSO:$H_2O_2$ (2:1:3 i.e. 15%) solution at room temperature for 12 to 24-h after methanol dehydration. Then, the samples were transferred back to 100% methanol and brought to −80°C three to five times for at least 1 h each time and back to room temperature to ensure that the antigens in the deeper parts of the tissue were rendered accessible.

## Whole-mount *in situ* hybridization
Whole-mount *in situ* HCR V3 was performed as previously described (Choi et al., 2018) using reagents from Molecular Instruments. In brief, samples for each condition were collected in 8-channel PCR tubes and incubated in 100 µl probe hybridization buffer for 5 min at room temperature and 30 min at 37°C before incubation with 1 pmol of each probe stock in 100 µl probe hybridization buffer overnight at 37°C. Next, samples were washed four times with 500 µl probe wash buffer for 15 min each wash at 37°C, twice with 1 ml 5× SSCT buffer for 10 min at room temperature and once with 100 µl amplification buffer for 30 min at room temperature. The hairpin mixture was prepared by separately heating 6 pmol of both h1 and h2 of each hairpin to 95°C for 90 s and incubating these at room temperature for 30 min in the dark. All the hairpin mixtures were then added to 100 µl amplification buffer, which was then added to the samples and incubated for 12-16 h at room temperature in the dark. Samples were then washed twice with a DAPI solution diluted in 5× SSCT buffer for 30 min each wash before imaging. Before each incubation in the sticky buffers, the solution was gently pipetted to ensure it was properly mixed. Finally, iPSM samples were mounted using Mount-G on a slide and covered by a coverglass prior to imaging. HCR probe design was as follows: *Uncx4.1* (accession NM 013702.3; hairpin B1); *Cdh2* (accession NM_007664.5, hairpin B1); *Mesp2* (accession NM_008589.2, hairpin B2); *Hes7* (accession NM_033041.5, hairpin B2); *Meox1* (accession NM_010791.3, hairpin B1); hairpin B1 was labelled with Alexa 647 and B2 with Alexa 594.

## Immunofluorescence staining
Immunostaining of iPSM was performed as previously described (Veenvliet et al., 2020). Following washes after methanol incubation, blocking and antibody incubations of the floating iPSM organoids were all carried out in 8-channel PCR tubes. Antibodies and dilutions used in this study were as follows: goat anti-Tbx6 (R&D Systems, AF4744; 1/200); rabbit phospho-p44/42 MAPK (Erk1/2) antibody (Cell Signaling Technology, 9101; 1/100), rabbit anti-Cdh2 (Cell Signaling Technology, 13116; 1/100) and guinea pig anti-Hes7 [home-made (Bessho et al., 2003); 1/50]. For detecting phosphorylated target protein, the handling of specific antibodies was adjusted by referring to the Cell Signaling Technology primary antibody product webpage where the methanol permeabilization step follows fixation. The cells were immersed or covered with ice–cold 100% methanol, incubated in methanol for 10 minutes at −20°C, rinsed in 1× PBS for 5 minutes. Secondary antibodies were conjugated to Alexa Fluor 594 for Tbx6 and Hes7, and Alexa Fluor 647 for Cdh2 and ppErk1/2. For staining with anti-Hes7 antibody, samples were washed with the blocking buffer (5% donkey serum/0.3%Triton/PBS) overnight between the primary and secondary antibody incubation to reduce nonspecific binding and background. Finally, the samples were mounted using Mount-G on a slide and covered by a coverglass prior to imaging. Confocal *z*-stack images were obtained on an LSM980 (ZEISS). Figures show average projection images.

## Statistical analysis
Results are presented as mean±s.e.m. In box-and-whiskers plots, the middle hinge corresponds to the median, the lower and upper hinges correspond to the first and third quartiles, respectively, and the lower and upper whiskers correspond to no further than the least and largest values, respectively. The analyses were visualized using R code (Engler, 2025). Unpaired two-tailed Student's *t*-test or ANOVA were used in this study: $P<0.05$ were considered significant. All experiments were performed in duplicate or triplicate.

## Acknowledgements
We thank Kotari Ishikawa and Hiromi Shimojo for technical support, Toshiyuki Ohtsuka, Taeko Kobayashi, Yuki Maeda and Ayumi Goto for discussion, Research Resources Division, RIKEN Center for Brain Science for peptide synthesis and other technical support, and Koichi Kawakami for the Tol2 transposon vector system. Some illustrations were generated using BioRender; see figure legends for details.

## Competing interests
The authors declare no competing or financial interests.

## Author contributions
Conceptualization: A.I., R.K.; Data curation: X.J.; Formal analysis: R.K; Funding acquisition: A.I., R.K.; Investigation: X.J.; Methodology: X.J., A.I.; Project administration: R.K.; Resources: A.I., R.K.; Supervision: R.K.; Visualization: X.J.; Writing – original draft: X.J., A.I., R.K.; Writing – review & editing: X.J., R.K.

## Funding
This work was funded by a Grant-in-Aid for Specially Promoted Research (21H04976 to R.K.) from the Japan Society for the Promotion of Science (JSPS); Precursory Research for Embryonic Science and Technology (JPMJPR2043 and JPMJPR15P1 to A.I.) from the Japan Science and Technology Agency (JST); a Grant-in-Aid for Scientific Research on Innovative Areas (19H04960 and 18H04734 to A.I.) and a Grant-in-Aid for Transformative Research Areas (A) (25H02478 to A.I.) from the Ministry of Education, Culture, Sports, Science, and Technology (MEXT), Japan; a Grant-in-Aid for Scientific Research (B) (21H03540 and 18H03332 to A.I.) from the JSPS. Open Access funding provided by RIKEN. Deposited in PMC for immediate release.

## Data and resource availability
ChIP-seq data have been deposited to Genbank under accession number PRJNA1294742. All other relevant data and details of resources can be found within the article and its supplementary information.

## The people behind the papers
This article has an associated 'The people behind the papers' interview with some of the authors.

## Peer review history
The peer review history is available online at https://journals.biologists.com/dev/lookup/doi/10.1242/dev.204743.reviewer-comments.pdf

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
