## [Peer Review File · Development (Cambridge, England)]

Cdh2, a downstream target of Hes7, regulates somitogenesis by supporting FGF signalling

Xueqi Jia, Akihiro Isomura and Ryoichiro Kageyama
DOI: 10.1242/dev.204743

Editor: Anna-Katerina Hadjantonakis

Review timeline

Original submission:	25 February 2025
Editorial decision:	5 April 2025
First revision received:	29 July 2025
Accepted:	14 August 2025

Original submission

First decision letter

MS ID#: dev.204743

MS TITLE: Cdh2, a downstream target of Hes7, regulates somitogenesis by supporting FGF signalling

AUTHORS: Xueqi Jia, Akihiro Isomura and Ryoichiro Kageyama

Dear Dr Kageyama,

I have now received all the referees' reports on the above manuscript, and have reached a decision. The referees' comments are appended below, or you can access them online: please go to:

As you will see, the referees express considerable interest in your work, but have some significant criticisms and recommend a substantial revision of your manuscript before we can consider publication. If you are able to revise the manuscript along the lines suggested, which may involve further experiments, I will be happy receive a revised version of the manuscript. Your revised paper will be re-reviewed by one or more of the original referees, and acceptance of your manuscript will depend on your addressing satisfactorily the reviewers' major concerns. Please also note that Development will normally permit only one round of major revision. If it would be helpful, you are welcome to contact us to discuss your revision in greater detail. Please send us a point-by-point response indicating your plans for addressing the referees' comments, and we will look over this and provide further guidance.

Please attend to all of the reviewers' comments and ensure that you clearly highlight all changes made in the revised manuscript. Please avoid using 'Tracked changes' in Word files as these are lost in PDF conversion. I should be grateful if you would also provide a point-by-point response detailing how you have dealt with the points raised by the reviewers in the 'Response to Reviewers' box. If you do not agree with any of their criticisms or suggestions please explain clearly why this is so.

Reviewer 1

Advance summary and potential significance to field

Somitogenesis is a complex process with inputs from many developmental signaling families. FGF and Notch have major impacts on this process. The authors present a very exciting connection

between FGF and Notch signaling. They show that *Cdh2* is regulated by *Hes7*, which is an essential regulator of the somitogenic Clock that is responsive to Notch signaling. Using Chip-seq, the authors identify *Cdh2* as a target gene of *HES7*. This Chip-seq data will be of high interest to the somitogenesis and Notch signaling fields. The authors then show that *Cdh2* loss of function iPSM organoids have abnormal clock oscillations and have reduced FGF signaling. The study offers a new insight into how the Clock may interact with FGF signaling during somitogenesis, which would be of wide interest to the field.

Comments for the author

The conclusions are for the most part supported by the data presented, however I feel that a few more pieces of data and clarifications will greatly increase the impact of the work. I have broken my comments into two categories, "Major" and "Minor" points.

Major Points

The authors demonstrate that *Cdh2* is transcriptional target of *Hes7* repression, and characterize the effect of loss of *Cdh2*, but it is unclear if *Hes7* regulation of *Cdh2* has an impact on somitogenesis and mechanistically how *Cdh2* affects clock oscillation. There is a lot of data presented within the current manuscript that would be useful to the somitogenesis field, and fully addressing the above two limitations is reasonably outside of the scope of this study, however the authors should address the two points below, which will greatly improve the paper.

1) The authors show that *Hes7* binds to the *Cdh2* promoter and propose that it is dynamically repressing *Cdh2* expression. I believe a little more work is required to show that periodic repression functions in somitogenesis. First, the data that *Cdh2* mRNA oscillates (Figure S6) should be strengthened. It is hard to distinguish an obvious oscillatory pattern of *Cdh2* mRNA expression in the images presented. Could the authors better show the oscillating expression by recoloring the channel with dynamic LUT color scheme such as "Fire"? Additionally, the graphs on the right side quantifying expression along the A-P axis have a line that looks like it denotes the anterior boundary of the PSM, the graphs should be labeled to this effect. The authors conclude that *Cdh2* protein does not oscillate but is in a posterior to anterior gradient. I believe that the authors explanation for this is that since the *Hes7* oscillation period is longer in the anterior PSM, this would lead to repression of *Cdh2* in the anterior PSM. Since the posterior period is shorter, this repression is less. If this is correct, it should be more emphasized and clearer in the manuscript. However if there is a gradient of protein, the mRNA pattern should reflect this same gradient, but it does not appear to do that. What is the explanation for this discrepancy? To test for the requirement of *Hes7* regulation of *Cdh2* in Clock function, deleting the *Hes7* binding site within the *Cdh2* promoter in iPSM cultures would be very clarifying. If *Hes7*-achilles oscillations are abnormal and *Cdh2* protein gradient is lost or reduced, this would prove the authors point. If the authors think this experiment is beyond the scope of this paper, they should list this lack of evidence for a direct connection between *Hes7* regulation and *Cdh2* function in the limitations section.

2) One possible explanation for how *Cdh2* is impacting the clock and somitogenesis is that *Cdh2* supports the wavefront activity which is required for Clock oscillation. From the authors data, it appears that *Cdh2* is required for wavefront signaling to maintain an undifferentiated PSM that is capable of transmitting clock oscillations without responding to the differentiation/segmentation signal. In Figure 2E&F there is an expansion of *Tbx6* in the *Cdh2*GOF iPSM, in Figure 3E the loss of ppERK in the posterior and the decrease in the size of the *Hes7*-achilles domain, and in Figure 4D the posterior shift of *Mesp2*, all suggest a change to the wavefront activity and/or position. Additionally, Figure S5B shows increased *Uncx4.1* in the anterior PSM as well as the posterior PSM, suggesting posterior shift of the differentiation front and premature differentiation of the PSM respectively. To address the possibility of loss of Wavefront activity and increased PSM differentiation, it would be helpful to examine the size of the *Tbx6* domain in *Cdh2* LOF iPSM and measure if expression is changed. To see if cells are differentiating prematurely, a somite differentiation marker (ie *Meox1*), could be used to see if expression is expanded in the anterior PSM or is present in the posterior PSM.

Minor Points

- Page 3, the term "clockwise" is confusing in the description of somite formation, as this term is typically used to denote a rotational direction, "sequential" seems to capture the point.

- Page3, it is not clear what the term "ticking" in reference to PSM cells is intended to mean.
- Figure 1E, in the example shown, Hes7-reporter oscillation in the Cdh2 ko iPSM tissue appears to be normal in the first ~half of imaging but becomes abnormal then stops oscillating. How reproducible is this pattern of loss in the Cdh2 ko? Could the authors report how many such embryos were imaged and what proportion showed loss and specifically if this pattern of normal early, lost late was observed?
- Pg 7, the sentence "However, in Cdh2-overexpressing iPSM tissue, pHes7-Achilles activity was expanded into a more anterior region (Fig. 2D and movie S3), suggesting that the more anterior cells still maintained PSM cell properties." I don't believe this panel addresses the anterior-posterior position of pHEs7-Achilles activity.
- Figure S4, there is no indication of statistical significance in the graph, is there an asterisk missing or is there no significant difference?
- Figure 4L, what do the two lines graphed and arrows pointing to either line indicate, variation/fluctuation? Please define in the figure legend or text.
- Figure S6, it is unclear if the images presented are single optical sections or subsets of a max intensity projection, please clarify in the figure legend.
- Throughout the paper mRNA names are inconsistently italicized. For clarity this should be consistent (my preference would be to always italicize mRNA).

Reviewer 2

Advance summary and potential significance to field

In this study, Jia et al. aimed to identify target genes of the core segmentation clock component Hes7 during somitogenesis. For this, they performed ChIP-seq and discovered Cdh2 as a new downstream target of Hes7. They also showed that it conversely also regulates Hes7 oscillations. They postulate this feedback loop involves a direct interaction between Cdh2 and the FGF receptor/ERK pathway, and perturbations of this Hes7/Cdh2 feedback loop resulted in disrupted timing of somite formation. This role for Cdh2 in the segmentation clock is novel and very interesting, and the experimental strategies used are elegant.

Comments for the author

However, some issues should be addressed before manuscript publication.

Major Comments

- The authors propose a model in which the loss of Cdh2 impacts PSM cell maturation through FGF signaling. However, an alternative model would be that loss of Cdh2 impacts differentiation and this in turn affects FGF signaling. We recommend adding this alternative interpretation in the discussion. Although it is probably out of the scope of this study, it would be nice if the authors can explore causality using a Cdh2 mutant that specifically disrupts the interaction with the FGFR.
- Furthermore, it is not clear how the Cdh2 protein can be part of the segmentation clock while the protein is uniformly expressed and as the authors conclude is not oscillatory itself. The authors should clarify this in their discussion.
- After showing Cdh2 is a direct downstream target of Hes7 by ChIP-seq, the authors have to validate it and ideally demonstrate whether this is an activating or inhibitory regulation. This has to be shown before they continue with their functional assessment of the interaction. Towards the end of the manuscript the effect a Hes7 KO on the Cdh2 levels is shown, which should be part of this initial validation.
- It is not obvious why the authors move from identifying downstream targets of Hes7 to studying the effect of these downstream targets on Hes7 itself. Why not investigate the effect on segmentation, especially with Cdh2 being one of the main targets? It should be justified better in the text that the focus is on identifying further components of the regulatory network and feedback mechanisms of the segmentation clock.

- The knockout should be validated. This includes sequencing data to show gene editing and confirm the absence of mRNA or protein expression (i.e. western blot or immunofluorescent staining).
- The specificity of the Hes7 ChIP-seq fully depends on the specificity of the used Hes7 antibody. The authors should validate that the antibody is specific for Hes7 and does not bind to other family members. Or refer to previous literature that validates this.
- The authors should confirm that iPSM differentiation works efficiently, even in Cdh2 knockout cells. Could a decrease in efficiency explain the effect on Hes7 oscillations or ppERK levels?
- Several statements about observations should be visualized properly and supported by quantifications:
 - o Please quantify tissue elongation and cell spreading for Cdh2 KO
 - o "The addition of doxycycline and exogenous protein overexpression led to weakened reporter signals in control and Cdh2-overexpressing iPSM tissue,..". This is neither shown nor quantified.
 - o 2A-D "... pHes7-Achilles activity in the anterior region was weakened after several oscillation cycles, ... However, in Cdh2-overexpressing iPSM tissue, pHes7-Achilles activity was expanded into a more anterior region (Fig. 2D and movie S3), ...". Kymographs should be added to show the spatial dynamics.
 - o 2D "overexpression of Cdh2 reduced amplitude of Hes7 oscillations": Please quantify the amplitude.
- The effect on cell-cell adhesion upon loss of Cdh2 is not sufficiently explored. Firstly, it is unclear to which extent the cell-cell contacts are affected, can the authors test this (e.g with showing the cell morphology, a-catenin localization...)? Furthermore, the authors state that the effect of Hes7 oscillations in Cdh2 KO iPSM is not due to lost cell-cell contact affecting Notch signalling because there is still internalized NICD. However, Hes7 can also be affected through other pathways than Notch that may be influenced by Cdh2 as well (e.g. Yap). To address this, the authors should more specifically disrupt the signaling function of Cdh2 without affecting adhesion, for instance with using an Cdh2 that lacks the intracellular domain, or restoring adhesion with another adhesion molecule.

Minor comments

- page 4. Radize et al. 1997 should be Radice et al. 1997
- please clarify the Hes7 reporter line, it is unclear how it is still reporting Hes7 dynamics in a Hes7 KO line.
- It is unclear in the results and figures (only in the methods) that overexpression of Cdh2 is done with the P2A system. The reason this is important is because the overexpression of Cdh2 may result in the downregulation of endogenous N-cad (as shown for E-Cad). However, in figure S4C it is shown that overexpression does result in elevated levels. This would be good to highlight more in the text.
- Please clarify why in Fig. 4D,F,H there was no switch to CLBF medium, what happens in that medium? Similarly, why are some experiments performed in Matrigel and others not?
- It would be good if the authors can add an example image for 4A and indicate herein the ROI's used to measure the periods in the posterior and anterior, respectively.

First revision

Author response to reviewers' comments

We would like to thank the Reviewers for their constructive comments. We answered their comments one by one as below. Reviewers' comments are in red, and our responses are in black.

To Reviewer #1:

Major Points

The authors demonstrate that *Cdh2* is transcriptional target of *Hes7* repression, and characterize the effect of loss of *Cdh2*, but it is unclear if *Hes7* regulation of *Cdh2* has an impact on somitogenesis and mechanistically how *Cdh2* affects clock oscillation. There is a lot of data presented within the current manuscript that would be useful to the somitogenesis field, and fully addressing the above two limitations is reasonably outside of the scope of this study, however the authors should address the two points below, which will greatly improve the paper.

1) The authors show that *Hes7* binds to the *Cdh2* promoter and propose that it is dynamically repressing *Cdh2* expression. I believe a little more work is required to show that periodic repression functions in somitogenesis. First, the data that *Cdh2* mRNA oscillates (Figure S6) should be strengthened. It is hard to distinguish an obvious oscillatory pattern of *Cdh2* mRNA expression in the images presented. Could the authors better show the oscillating expression by recoloring the channel with dynamic LUT color scheme such as "Fire"? Additionally, the graphs on the right side quantifying expression along the A-P axis have a line that looks like it denotes the anterior boundary of the PSM, the graphs should be labeled to this effect. The authors conclude that *Cdh2* protein does not oscillate but is in a posterior to anterior gradient. I believe that the authors explanation for this is that since the *Hes7* oscillation period is longer in the anterior PSM, this would lead to repression of *Cdh2* in the anterior PSM. Since the posterior period is shorter, this repression is less. If this is correct, it should be more emphasized and clearer in the manuscript. However if there is a gradient of protein, the mRNA pattern should reflect this same gradient, but it does not appear to do that. What is the explanation for this discrepancy? To test for the requirement of *Hes7* regulation of *Cdh2* in Clock function, deleting the *Hes7* binding site within the *Cdh2* promoter in iPSM cultures would be very clarifying. If *Hes7*-achilles oscillations are abnormal and *Cdh2* protein gradient is lost or reduced, this would prove the authors point. If the authors think this experiment is beyond the scope of this paper, they should list this lack of evidence for a direct connection between *Hes7* regulation and *Cdh2* function in the limitations section.

We thank this reviewer for his/her insightful suggestions and agree that it is interesting to understanding how exactly the regulation of *Hes7* on *Cdh2* would involve in the somitogenesis. We have included additional experiments to provide as much as information to understand the process.

Following the reviewer's advice, we adjusted the figure to fire to have another way for presentation (Fig. S5A). In addition, the anterior boundary is indicated in the graph (Fig. S5A). There seems to have a week dynamics for the mRNA expression. However the protein distribution is mostly showing a steady gradient (Fig. S5B). So, we generated a cell line with mScarlet knocked in to the C terminus of the *Cdh2* coding sequence so that *Cdh2*-mScarlet fusion protein is expressed. Live imaging by confocal microscopy showed that the tracing of the population intensity of the reporter had small fluctuations along time while no obvious wave or "on and off" pattern can be observed (Fig. 3A-D and movie S3). When measuring the intensity along the posterior to anterior iPSM, a decreasing gradient was observed, as shown in the mouse PSM immunostaining (Fig. S5B). Therefore, it is possible that the protein level of *Cdh2* is not oscillating but gradually decreases as cells differentiate. To explain why the patterns between mRNA and protein are different, we showed that the half life of *Cdh2* protein is approximately 3 h (Fig. 3E-G), which is much longer than that of known clock genes, e.g. about 22 min for *Hes7* protein (Hirata et al., 2004). Considering also the mild dynamic mRNA expression patterns, it is reasonable that *Cdh2* protein forms a steady gradient.

We also discussed in Study limitations that the direct relationship between *Hes7* binding to the *Cdh2* locus and segmentation clock function has not yet been addressed, as we have not specifically removed the binding site.

2) One possible explanation for how *Cdh2* is impacting the clock and somitogenesis is that *Cdh2* supports the wavefront activity which is required for Clock oscillation. From the authors data,

it appears that *Cdh2* is required for wavefront signaling to maintain an undifferentiated PSM that is capable of transmitting clock oscillations without responding to the differentiation/segmentation signal. In Figure 2E&F there is an expansion of *Tbx6* in the *Cdh2*GOF iPSM, in Figure 3E the loss of ppERK in the posterior and the decrease in the size of the *Hes7*-achilles domain, and in Figure 4D the posterior shift of *Mesp2*, all suggest a change to the wavefront activity and/or position. Additionally, Figure S5B shows increased *Uncx4.1* in the anterior PSM as well as the posterior PSM, suggesting posterior shift of the differentiation front and premature differentiation of the PSM respectively. To address the possibility of loss of Wavefront activity and increased PSM differentiation, it would be helpful to examine the size of the *Tbx6* domain in *Cdh2* LOF iPSM and measure if expression is changed. To see if cells are differentiating prematurely, a somite differentiation marker (ie *Meox1*), could be used to see if expression is expanded in the anterior PSM or is present in the posterior PSM.

To collect more evidence to support the possible shifted wavefront activity, we quantified the *TBX6* distribution in case of *Cdh2* KO iPSMs as this reviewer suggested. The measurement of *Tbx6* intensity along the iPSMs showed there is a tendency of posterior shift in *Cdh2* KO iPSMs compared to control (Figs. 4E-G), indicating the increased differentiation potential in the *Cdh2*KO iPSM. We also found that *Moex1* expression occurred in the PSM region when *Cdh2* was absent (Fig. S9D).

Minor Points

- Page 3, the term "clockwise" is confusing in the description of somite formation, as this term is typically used to denote a rotational direction, "sequential" seems to capture the point.

We thank the reviewer for this suggestion, and it is now corrected in the text.

- Page3, it is not clear what the term "ticking" in reference to PSM cells is intended to mean. We deleted this word.

- Figure 1E, in the example shown, *Hes7*-reporter oscillation in the *Cdh2* ko iPSM tissue appears to be normal in the first -half of imaging but becomes abnormal then stops oscillating. How reproducible is this pattern of loss in the *Cdh2* ko? Could the authors report how many such embryos were imaged and what proportion showed loss and specifically if this pattern of normal early, lost late was observed?

We generated 3 independent cell lines and performed live-cell imaging at least 3 times for each cell line. We described it on page 19 (lines 7-8) and page 21 (lines 9-10).

- Pg 7, the sentence "However, in *Cdh2*-overexpressing iPSM tissue, p*Hes7*-Achilles activity was expanded into a more anterior region (Fig. 2D and movie S3), suggesting that the more anterior cells still maintained PSM cell properties." I don't believe this panel addresses the anterior-posterior position of p*Hes7*-Achilles activity.

New data concerning this point is now presented in new Fig. 4D (lower graph). This anterior ROI was placed at the same distance from the posterior end. In the control (-Dox), *Hes7* oscillations were dampened because this region probably differentiated into somites whereas in *Cdh2*-overexpressing iPSM (+Dox), *Hes7* oscillations still continued (Fig. 4D). These results indicate that *Cdh2* overexpression may delay the exit from PSM property.

- Figure S4, there is no indication of statistical significance in the graph, is there an asterisk missing or is there no significant difference?

This figure is now Fig. S7, and we added the statistical indication to the graphs.

- Figure 4L, what do the two lines graphed and arrows pointing to either line indicate, variation/fluctuation? Please define in the figure legend or text.

The magenta lines represent the expression intensity along the axis, and the black arrow headed lines between the magenta lines indicate the fluctuations between samples. We have modified the figure legend for better understanding.

- Figure S6, it is unclear if the images presented are single optical sections or subsets of a max intensity projection, please clarify in the figure legend.

This is now new Fig. S5. All the images are projected imaged from a z-stack. We now modified the figure legend for better understanding.

- Throughout the paper mRNA names are inconsistently italicized. For clarity this should be consistent (my preference would be to always italicize mRNA).

We thank the reviewer for this suggestion, and we made all mRNA names italicized.

To Reviewer #2:

Major Comments

- The authors propose a model in which the loss of Cdh2 impacts PSM cell maturation through FGF signaling. However, an alternative model would be that loss of Cdh2 impacts differentiation and this in turn affects FGF signaling. We recommend adding this alternative interpretation in the discussion. Although it is probably out of the scope of this study, it would be nice if the authors can explore causality using a Cdh2 mutant that specifically disrupts the interaction with the FGFR.

We thank the reviewer for this suggestion and agreed that it would be more direct to explore the relation between Cdh2 and FGFR without affecting other factors. So, we performed two experiments to address this issue. Previous studies examined the physical interaction between Cdh2 and FGFR1 protein and identified the sequence in the extracellular domain 4 of Cdh2, which is responsible for interaction with FGFR1 (Williams et al., 2001). Taking advantage of this information, we examined the effects on PSM cell differentiation by only interfering the interaction of the extracellular domain 4 of Cdh2 with FGFR1 by either applying the blocking peptide containing this site or deleting this extracellular domain from Cdh2 (Fig. S8). We found that in both cases, FGF-ppERK signaling was downregulated, shortening the TBX6 positive proportion in the iPSM (Fig. S8). These data indicate that blocking the interaction between Cdh2 and FGFR1 reduced FGF signaling activity and affected the PSM cell differentiation.

- Furthermore, it is not clear how the Cdh2 protein can be part of the segmentation clock while the protein is uniformly expressed and as the authors conclude is not oscillatory itself. The authors should clarify this in their discussion.

We thank the reviewer for this comment, and we agree that Cdh2 itself may not be a clock gene oscillating in the PSM cells, while the pulsatile repressions by Hes7 may fine-tune Cdh2 protein levels or gradients. We generated Cdh2-mScarlet knock-in ES cells, which were induced to become iPSM, and we confirmed that Cdh2 protein formed a rather steady gradient (new Fig. 3). We found that Hes7-controlled Cdh2 gradient regulates FGF signalling and ppERK oscillations, which are the key components in the segmentation clock. We discussed that this Cdh2 gradient is required for proper maintenance of Hes7 oscillations, as loss of Cdh2 prematurely stops Hes7 oscillations while Cdh2 overexpression prolongs Hes7 oscillations. In this regard, Cdh2 is a key component of proper maintenance of the segmentation clock.

- After showing Cdh2 is a direct downstream target of Hes7 by ChIP-seq, the authors have to validate it and ideally demonstrate whether this is an activating or inhibitory regulation. This has to be shown before they continue with their functional assessment of the interaction. Towards the end of the manuscript the effect a Hes7 KO on the Cdh2 levels is shown, which should be part of this initial validation.

We thank the reviewer for this suggestion, and we now presented the validation data in new Fig. 2.

- It is not obvious why the authors move from identifying downstream targets of Hes7 to studying the effect of these downstream targets on Hes7 itself. Why not investigate the effect on

segmentation, especially with Cdh2 being one of the main targets? It should be justified better in the text that the focus is on identifying further components of the regulatory network and feedback mechanisms of the segmentation clock.

According to this reviewer's comment, we stated the following content in the introduction. Hes7- induced oscillatory networks have been well characterized, while the full regulatory mechanisms and downstream effectors remain incompletely understood. So, to clarify the regulatory networks of Hes7, we conducted ChIP-seq analysis using Hes7 antibody, expecting some of the Hes7 target genes regulate Hes7 oscillations, like Lfng. This is why we introduced mutations into Hes7 target genes in ES cells carrying a Hes7 reporter and performed live imaging in iPSM. However, as the reviewer has pointed out, we do not have much observations on somites formation for the targets, which we discussed briefly in the limitation session.

- The knockout should be validated. This includes sequencing data to show gene editing and confirm the absence of mRNA or protein expression (i.e. western blot or immunofluorescent staining).

The sequence data were added as Figure S3 with 2 or 3 clones having frameshift mutations as presented. The absence of protein expression of Cdh2 has been confirmed by the western blotting (Fig. 5B).

- The specificity of the Hes7 ChIP-seq fully depends on the specificity of the used Hes7 antibody. The authors should validate that the antibody is specific for Hes7 and does not bind to other family members. Or refer to previous literature that validates this.

Hes7 antibody used in this study has been used for many previous studies (Bessho et al. 2003 Genes Dev; Hirata et al. 2004 Nat Genet; Niwa et al. 2011 Genes Dev, etc.), and its specificity has been confirmed. We also used it for immunostaining of Hes7 in this study (Figs. 1A and S5B). It only detected Hes7 signals in the PSM but was not reactive to Hes1 or Hes5 in the nervous system.

- The authors should confirm that iPSM differentiation works efficiently, even in Cdh2 knockout cells. Could a decrease in efficiency explain the effect on Hes7 oscillations or ppERK levels?

We showed that Hes7 oscillations and the PSM marker TBX6 expression occurred normally in the iPSM at least during the initial 20 h. This suggests that Cdh2 KO iPSM can be induced efficiently.

- Several statements about observations should be visualized properly and supported by quantifications:

Please quantify tissue elongation and cell spreading for Cdh2 KO

We now showed the data in Fig. S8C,D and Fig. S8I.

"The addition of doxycycline and exogenous protein overexpression led to weakened reporter signals in control and Cdh2-overexpressing iPSM tissue,..". This is neither shown nor quantified.

To answer this comment, we showed the data in Fig. S6G.

2A-D "... pHes7-Achilles activity in th4e anterior region was weakened after several oscillation cycles, ... However, in Cdh2-overexpressing iPSM tissue, pHes7-Achilles activity was expanded into a more anterior region (Fig. 2D and movie S3), ...". Kymographs should be added to show the spatial dynamics..

We added kymographs in Fig. S6E,F.

2D "overexpression of Cdh2 reduced amplitude of Hes7 oscillations": Please quantify the amplitude.

We now noticed that the previously stated "reduced amplitude" is due to the average of

different ROIs. There is no difference observed in comparing the single ROI plot (Fig. 4D). We corrected it in the main text.

- The effect on cell-cell adhesion upon loss of Cdh2 is not sufficiently explored. Firstly, it is unclear to which extent the cell-cell contacts are affected, can the authors test this (e.g with showing the cell morphology, a-catenin localization...)? Furthermore, the authors state that the effect of Hes7 oscillations in Cdh2 KO iPSM is not due to lost cell-cell contact affecting Notch signalling because there is still internalized NICD. However, Hes7 can also be affected through other pathways than Notch that may be influenced by Cdh2 as well (e.g. Yap). To address this, the authors should more specifically disrupt the signaling function of Cdh2 without affecting adhesion, for instance with using an Cdh2 that lacks the intracellular domain, or restoring adhesion with another adhesion molecule.

This is a really interesting point and inspired us to explore this part with additional experiments. As stated above to answer Reviewer 1, we performed two additional experiments to address this issue. Previous studies examined the physical interaction between Cdh2 and FGFR1 protein and identified the sequence in the extracellular domain 4 of Cdh2, which is responsible for interaction with FGFR1 (Williams et al., 2001). Taking advantage of this information, we examined the effects on PSM cell differentiation by only interfering the interaction of the extracellular domain 4 of Cdh2 with FGFR1 by either applying the blocking peptide containing this site or deleting this extracellular domain from Cdh2. We found that in both cases, FGF-ppERK signaling was downregulated, shortening the TBX6 positive proportion in the iPSM (Fig. S8). These data indicate that blocking the interaction between Cdh2 and FGFR1 reduced FGF signaling activity and affected the PSM cell differentiation.

Minor comments

- page 4. Radize et al. 1997 should be Radice et al. 1997

The text is corrected.

- please clarify the Hes7 reporter line, it is unclear how it is still reporting Hes7 dynamics in a Hes7 KO line.

The Hes7 reporter cell line used in this study is the same as described previously (Isomuta et al. 2024). A construction of pHes7-Achilles is integrated into the ES genome by Tol2 system. Therefore, when Hes7 protein is removed, there is no negative feedback, allowing the constant expression of the pHes7-Achilles reporter, as shown in the movie (Movie.S1) and the measurement (Fig. 1).

- It is unclear in the results and figures (only in the methods) that overexpression of Cdh2 is done with the P2A system. The reason this is important is because the overexpression of Cdh2 may result in the downregulation of endogenous N-cad (as shown for E-Cad). However, in figure S4C it is shown that overexpression does result in elevated levels. This would be good to highlight more in the text.

The overexpression of Cdh2 is done using a construct carrying Cdh2 coding sequence followed by a P2A sequence and mCherry coding sequence which is indicated in the schematic figure (Fig. 4A). We also stated that Tet-ON-inducible overexpression of Cdh2 was confirmed by western blotting (see Fig. S7C) on page 8 (lines 4-5 in the second paragraph).

- Please clarify why in Fig. 4D,F,H there was no switch to CLBF medium, what happens in that medium? Similarly, why are some experiments performed in Matrigel and others not?

In this study, we used different medium according to the purpose of the experiments. CL medium is the PSM induction medium that is critical for proper iPSM formation (Isomura et al., 2024). The CLBF medium is the CL medium supplemented with bFGF and an inhibitor for RA signalling to keep the cells in a undifferentiated state for a longer time which enhances the posterior state of iPSM property for prolonged periods. This is stated on page 20 (line 7). We found this condition is optimal for imaging and detecting noises to synchronicity, however it can add artifacts in the differentiation process, so we kept cells in the CL medium in the

experiments for exploring FGF activity and differentiation. In case of the experiments for checking somites properties, we used Matrigel condition to recapture a more natural state but not for other experiments due to the difficulty for handling and live imaging of the iPSM. We found that it is useful to apply different conditions for different purposes.

- It would be good if the authors can add an example image for 4A and indicate herein the ROI's used to measure the periods in the posterior and anterior, respectively.

This is now new Fig. 6A, and we added ROIs accordingly.

Second decision letter

MS ID#: dev.204743R1

MS TITLE: Cdh2, a downstream target of Hes7, regulates somitogenesis by supporting FGF signalling

AUTHORS: Xueqi Jia, Akihiro Isomura and Ryoichiro Kageyama

Dear Dr Kageyama,

I am happy to tell you that your manuscript has been accepted for publication in Development, pending our standard publication integrity checks.

Reviewer 1

Advance summary and potential significance to field

This manuscript by Jia et presents a novel insight into the complex relationship between the somitogenic clock and the processes regulating somite differentiation. They show that Hes7, an essential regulator of the clock, also regulates Cdh2 expression which in turn impacts somite differentiation involving Fgf signaling. This work provides high impact data and insights on these processes. Additionally, the Hes7 ChIP-seq data is very valuable for researchers studying gene regulation networks related to Hes7. The authors have completely addressed all of my initial concerns and their efforts are much appreciated.

Reviewer 2

Advance summary and potential significance to field

Our comments have been addressed in detail and further experiments were added to support this. No further comments.